# LLM-TS Integrator: Integrating LLM for Enhanced Time Series Modeling

**Can (Sam) Chen** *                                                     *can.chen@mila.quebec*
*Mila - Quebec AI Institute*

**Gabriel L. Oliveira**                                                  *gabrielleivas@gmail.com*
*RBC Borealis*

**Hossein Sharifi-Noghabi**                                             *hossein.sharifi@borealisai.com*
*RBC Borealis*

**Tristan Sylvain**                                                     *tristan.sylvain@borealisai.com*
*RBC Borealis*

**Reviewed on OpenReview:** *https://openreview.net/forum?id=vPVqQmjCy8*

## Abstract

Time series (TS) modeling is essential in dynamic systems like weather prediction and anomaly detection. Recent studies utilize Large Language Models (LLMs) for TS modeling, leveraging their powerful pattern recognition capabilities. These methods primarily position LLMs as the predictive backbone, often omitting the mathematical modeling within traditional TS models, such as periodicity. However, disregarding the potential of LLMs also overlooks their pattern recognition capabilities. To address this gap, we introduce *LLM-TS Integrator*, a novel framework that effectively integrates the capabilities of LLMs into traditional TS modeling. Central to this integration is our *mutual information* module. The core of this *mutual information* module is a traditional TS model enhanced with LLM-derived insights for improved predictive abilities. This enhancement is achieved by maximizing the mutual information between traditional model's TS representations and LLM's textual representation counterparts, bridging the two modalities. Moreover, we recognize that samples vary in importance for two losses: traditional prediction and mutual information maximization. To address this variability, we introduce the *sample reweighting* module to improve information utilization. This module assigns dual weights to each sample: one for prediction loss and another for mutual information loss, dynamically optimizing these weights via bi-level optimization. Our method achieves state-of-the-art or comparable performance across five mainstream TS tasks, including short-term and long-term forecasting, imputation, classification, and anomaly detection. Our code is available at: `https://github.com/BorealisAI/LLM-TS-Integrator`

## 1 Introduction

Time series (TS) modeling, as emphasized in Hyndman and Athanasopoulos (2018), is crucial for a variety of real-world applications. It is instrumental in forecasting meteorological factors for weather prediction (Wu et al., 2021), imputing missing data in economic TS (Friedman, 1962), detecting anomalies in industrial monitoring data for maintenance (Gao et al., 2020), and classifying trajectories for action recognition (Franceschi et al., 2019). Given its significant practical impact, TS analysis continues to attract substantial attention (Lim and Zohren, 2021; Wen et al., 2022).

---

*Work done during an internship at RBC Borealis.

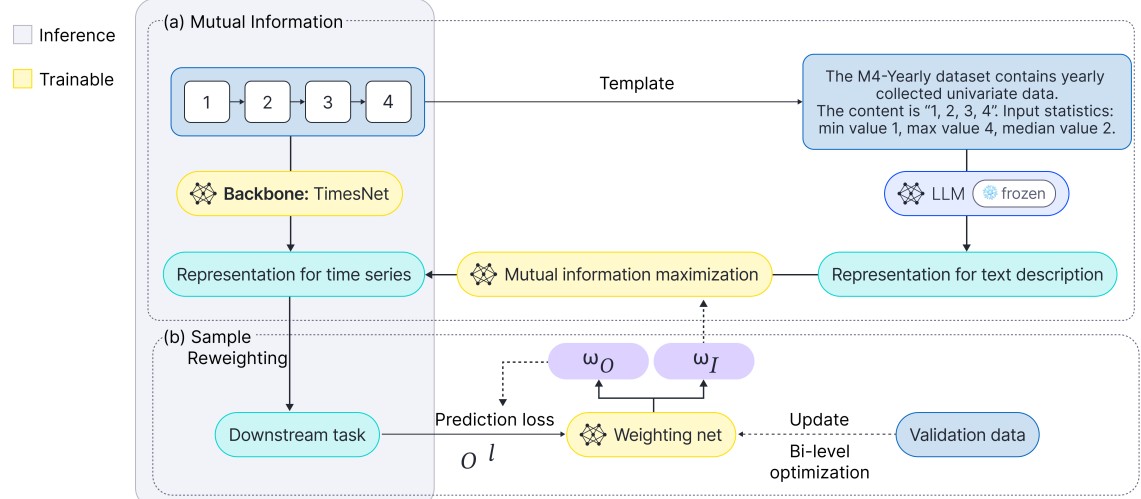

Figure 1: Illustration of *LLM-TS Integrator*. Module (a) enhances the traditional TS model (TimesNet) with LLM-derived insights by mutual information maximization. Module (b) optimizes sample importance for both prediction loss and mutual information loss to improve information utilization. The LLM is utilized solely during the training phase and is not required during inference.

Recent efforts in TS modeling have increasingly adopted Large Language Models (LLMs) to leverage their exceptional pattern recognition capabilities (Jiang et al., 2024; Zhou et al., 2023; Jin et al., 2024; Sun et al., 2023; Gruver et al., 2023b; Cao et al., 2024). While these innovative approaches validate the potential of LLMs in TS modeling, they primarily position LLMs as the core predictive model. Consequently, they often omit the mathematical modeling tailored specifically for TS models, such as employing the Fourier Transform to capture periodic patterns (Wu et al., 2023).

On the other hand, fully disregarding the potential of LLMs also overlooks their powerful pattern recognition capabilities. It is important to recognize the balance between leveraging LLMs for their advanced capabilities and utilizing traditional TS models for their mathematical modeling, to enhance the overall performance and accuracy of TS predictions. In response, we propose *LLM-TS Integrator*, a novel framework that effectively integrates the capabilities of LLMs into traditional TS modeling.

Central to our framework is a *mutual information* module, as depicted in Figure 1(a). The core of this module is a traditional predictive model, which we enhance with insights derived from LLMs to improve its predictive abilities. In this work, we primarily utilize TimesNet (Wu et al., 2023) as the traditional predictive model due to its exceptional performance and insight into periodic modeling, and our framework is also applicable to other traditional TS models (see in Section 4.6.3). We achieve this enhancement by maximizing the mutual information (Sun et al., 2020) between the TS representations from traditional models and their textual counterparts from LLMs, thereby bridging these two modalities (a detailed discussion of various LLMs is provided in Section 4.6.4). Despite its established use, mutual information maximization has not been previously applied to the intersection of TS and text domain. With textual descriptions often missing from TS data, we propose generating such descriptions via a carefully designed template. This template is enriched with essential background and statistical details pertinent to the TS, thereby enriching the LLM's comprehension of the TS context (a comprehensive discussion of various templates is in Appendix A.5).

Our first module introduces a dual loss framework: traditional prediction and mutual information, and we recognize that the importance of samples differs between the two losses. For instance, a large prediction loss for a sample highlights its learning potential, emphasizing the need to focus on its prediction loss. This scenario also implies that the model's learning for this sample is inadequate and its hidden representation is suboptimal for mutual information computation. Consequently, the sample's contribution to the mutual information calculation should be reduced. To manage this variability, we have introduced a novel *sample*

*reweighting* module powered by a simple MLP (multilayer perceptron) network, as depicted in Figure 1(b). This module processes the sample prediction loss to produce dual weights for each sample, one for the prediction loss and another for the mutual information loss. These weights are optimized through bi-level optimization, thereby enhancing the efficacy of information utilization.

Our primary contributions are as follows:

- We introduce the *LLM-TS Integrator* framework, which consists of *mutual information* and *sample reweighting*. The first module enhances traditional TS modeling with capabilities from LLMs through mutual information maximization.

- The second module optimizes sample importance for both prediction loss and mutual information loss, which improves information utilization.

- Extensive experiments across five mainstream TS tasks—short-term and long-term forecasting, imputation, classification, and anomaly detection—demonstrate the effectiveness of our framework. Obtained experimental results align across state-of-the-art of traditional time series and various LLMs which demonstrates applicability of *LLM-TS Integrator* framework regardless of choice of methods.

- *LLM-TS Integrator* framework keeps LLMs frozen and does not require fine-tuning and introduces minimum additional costs compared to traditional time series methods.

## 2    Preliminaries

**TimesNet.** In this paper, we mainly choose TimesNet as the traditional predictive model due to its exceptional performance (Wu et al., 2023) and also explore other additional traditional models including ETSformer (Woo et al., 2022), Stationary (Liu et al., 2022b), and FreTS (Yi et al., 2023) in Section 4.6. Previous studies to modeling temporal variations in 1D time series often struggle with complex temporal patterns. TimesNet addresses this challenge by decomposing these complex variations into multiple intra-period and inter-period variations. This is achieved by transforming the 1D time series into a series of 2D tensors, each corresponding to different periods. For the time series $\boldsymbol{x}$, we derive its representation $\boldsymbol{h}_{\boldsymbol{\theta}}^m(\boldsymbol{x})$ using the TimesNet model parameterized by $\boldsymbol{\theta}$ where $m$ represents *model*.

**Large Language Models.** Language models are trained on extensive collections of natural language sequences, with each sequence consisting of multiple tokens. Notable large language models such as GPT-3 (Brown et al., 2020) and Llama2 (Touvron et al., 2023) aim to predict the next token based on preceding tokens, demonstrating their capabilities through improvements in model parameter size and the amount of training data. Each language model uses a tokenizer that breaks down an input string into a sequence of recognizable tokens. However, the training of current large language models is solely focused on natural language, not encompassing time series data. This limitation presents challenges for the direct application of large language models to time series analysis.

## 3    Method

In this section, we present the *LLM-TS Integrator* framework, which effectively integrates the capabilities of LLMs into traditional TS modeling. This framework consists of two modules: *mutual information* and *sample reweighting*. The first module enhances a traditional TS model with LLM-derived insights for improved predictive abilities, as explored in Section 3.1. The second module optimizes weights for prediction loss and mutual information loss via bi-level optimization, improving information utilization, as covered in Section 3.2. The overall algorithm is shown in Algorithm 1.

### 3.1    Mutual Information

Previous studies (Zhou et al., 2023; Jin et al., 2023) have predominantly highlighted the use of Large Language Models (LLMs) as the core predictive model in the TS analysis, often omitting the mathematical modeling within traditional TS models, such as periodicity.

---

**Algorithm 1 LLM-TS Integrator**

---

**Input**: The TS dataset $\mathcal{D}$, number of training iterations $T$.

**Output**: Trained TS model parameterized by $\boldsymbol{\theta}^*$.

1: /* *Mutual Information Module* */
2: Train a traditional TS model (e.g., TimesNet) parameterized by $\boldsymbol{\theta}$ using $\mathcal{D}$.
3: Generate text description $\boldsymbol{t}$ for TS sample $\boldsymbol{x}$ via a designed template.
4: Derive hidden representations $\boldsymbol{h}_{\boldsymbol{\theta}}^m(\boldsymbol{x})$ from the TS model and $\boldsymbol{h}^l(\boldsymbol{t})$ from the LLM.
5: **while** $\tau <= T - 1$ **do**
6:     Sample $\boldsymbol{x}$, $\boldsymbol{t}$, $\boldsymbol{y}$ from $\mathcal{D}$, where $\boldsymbol{y}$ are the labels.
7:     Optimize a discriminator model $T_{\boldsymbol{\beta}}$ to estimate mutual information as per Eq .(2).
8:     /* *Sample Reweighting Module* */
9:     Process sample loss $l_O$ with the weighting net to produce dual weights as per Eq. (3), (4).
10:     Adopt bi-level optimization to update the weighting net following Eq. (6), (7).
11:     Re-calculate dual weights using the updated weighting net per Eq. (3), (4).
12:     Calculate the overall loss to update the TS model as per Eq. (5).
13: **end while**
14: Return the trained TS model parameterized by $\boldsymbol{\theta}^*$.

---

In contrast, our framework utilizes a traditional TS model as the predictive backbone, enhanced by the advanced capabilities of LLMs. In this paper, we employ TimesNet, as outlined in Section 2, as the traditional predictive model, and we further examine other models in Section 4.6. This hybrid methodology combines the advantages of both traditional TS models and modern LLMs. We achieve this integration via a *mutual information* module, which maximizes the mutual information between the TS data representations derived from the traditional model and their corresponding textual representations derived from LLMs.

### 3.1.1 Mutual Information Estimation.

Estimating the mutual information between hidden representations of a time series (TS) sample $\boldsymbol{x}$ and its corresponding textual description $\boldsymbol{t}$ is essential. For the TS sample $\boldsymbol{x}$, we derive its representation $\boldsymbol{h}_{\boldsymbol{\theta}}^m(\boldsymbol{x})$ using TimesNet, a **m**odel parameterized by $\boldsymbol{\theta}$. For the text $\boldsymbol{t}$, its representation $\boldsymbol{h}^l(\boldsymbol{t})$ is extracted using a pre-trained LLM, where $l$ denotes the language model. In this study, we employ the LLaMA-3b model (Touvron et al., 2023) as our primary LLM, while also evaluating other LLMs as detailed in Section 4.6.

We estimate mutual information using the Jensen-Shannon MI estimator (Sun et al., 2020; Nowozin et al., 2016) and additionally explore the MINE estimator (Hjelm et al., 2019a) as detailed in Appendix A.12. Specifically, let $(\boldsymbol{x}, \boldsymbol{t})$ represent a sample from the TS set $\mathbb{S}$, and $(\tilde{\boldsymbol{x}}, \tilde{\boldsymbol{t}})$ denote a sample from $\tilde{\mathbb{S}} = \mathbb{S}$ where $(\boldsymbol{x}, \boldsymbol{t}) \neq (\tilde{\boldsymbol{x}}, \tilde{\boldsymbol{t}})$. Within this context, $\mathbb{S}$ denotes the TS training distribution while the product $\mathbb{S} \times \tilde{\mathbb{S}}$ represents pairs of distinct samples within $\mathbb{S}$. Then the lower bound of mutual information can be estimated as:

$$I(\boldsymbol{\theta}, \boldsymbol{\beta}) = \mathbb{E}_{\mathbb{S}}[-sp(-T_{\boldsymbol{\beta}}(\boldsymbol{h}_{\boldsymbol{\theta}}^m(\boldsymbol{x}), \boldsymbol{h}^l(\boldsymbol{t})))] - \mathbb{E}_{\mathbb{S} \times \tilde{\mathbb{S}}}[sp(T_{\boldsymbol{\beta}}(\boldsymbol{h}_{\boldsymbol{\theta}}^m(\boldsymbol{x}), \boldsymbol{h}^l(\tilde{\boldsymbol{t}})))] \,, \tag{1}$$

where $T_{\boldsymbol{\beta}}$ denotes the discriminator parameterized by $\boldsymbol{\beta}$ and $sp$ is the softplus function. For the details of $T_{\boldsymbol{\beta}}$, we feed the positive and negative examples into a 1-layered fully-connected network with a hidden size of 64, and then output the dotproduct of the two representations. The mutual information estimation begins by fixing the model parameters $\boldsymbol{\theta}$, followed by training $\boldsymbol{\beta}$ as the estimator. Specifically, we optimize $\boldsymbol{\beta}$ via the following:

$$\hat{\boldsymbol{\beta}} = \boldsymbol{\beta} - \eta_0 \cdot \frac{\partial I(\boldsymbol{\theta}, \boldsymbol{\beta})}{\partial \boldsymbol{\beta}} \tag{2}$$

where $\eta_0$ denotes the learning rate. Subsequently, we refine the model parameters $\boldsymbol{\theta}$ to maximize mutual information, thereby enriching the traditional TS model with insights derived from LLMs. This alternating optimization procedure between model and discriminator is repeated each epoch.

### 3.1.2 Text Description for Time Series.

In our approach, we assume each time series (TS) sample $\boldsymbol{x}$ is paired with a corresponding textual description, $\boldsymbol{t}$. However, textual descriptions are frequently unavailable for many TS datasets. To bridge this gap, we introduce a methodology for generating textual descriptions of TS data. We propose creating textual representations that capture essential background and statistical details of the TS inspired by Jin et al. (2023). This process can be systematically implemented using the following carefully designed template:

```
template = (
    "{task_description}. The content is: {TS}. "
    "Input statistics: min value {min(TS)}, max value {max(TS)}, "
    "median value {median(TS)}, top 5 lags {compute_lags(TS)}."
)
```

## 3.2 Sample Reweighting

Our *mutual information* module introduces two distinct loss functions: (1) the original sample prediction loss, $l_O(\boldsymbol{x}, \boldsymbol{y})$, hereafter referred to as $l_O$, which corresponds to the prediction loss for a TS sample $\boldsymbol{x}$ and its label $\boldsymbol{y}$, and (2) the mutual information maximization loss, denoted as $-I(\boldsymbol{\theta}, \boldsymbol{\beta})$. We acknowledge that the significance of samples varies between these two losses. Specifically, a large prediction loss $l_O$ indicates a sample's substantial learning potential, thereby justifying a higher weight $\omega_O$ for its prediction loss. Conversely, this suggests that the sample's representation may be suboptimal for mutual information computation, warranting a lower weight $\omega_I$.

Here, $l_O(\boldsymbol{x}, \boldsymbol{y})$ denotes the sample-level loss for each task: For the forecasting task, we use MSE, where $\boldsymbol{y}$ represents the future target values. For the classification task, we use cross-entropy, where $\boldsymbol{y}$ corresponds to a class label. For the imputation task, we compute MSE for the masked points that need to be reconstructed. For the anomaly detection task, we adopt a reconstruction-based approach, measuring MSE between the original and reconstructed time series.

To address this disparity, we have developed a novel *sample reweighting* module, described as follows.

### 3.2.1 Weighting Network.

To automate weight assignment, our module employs a two-layer MLP network parameterized by $\boldsymbol{\alpha}$, which processes the sample prediction loss to produce a pair of weights:

$$\omega_O(\boldsymbol{\alpha}), \omega_I(\boldsymbol{\alpha}) = MLP_{\boldsymbol{\alpha}}(l_O) \tag{3}$$

This process involves converting the sample loss $l_O$ into a latent code $z$ through a hidden layer. The network then outputs dual weights:

$$\omega_O(\boldsymbol{\alpha}), \omega_I(\boldsymbol{\alpha}) = \sigma(m_O \cdot z), \sigma(m_I \cdot z) \tag{4}$$

where $m_O > 0$ and $m_I < 0$ to ensure a negative correlation between $\omega_O$ and $\omega_I$. The function $\sigma(\cdot)$ denotes the sigmoid activation.

For a batch of $N$ samples, the weight vector $\boldsymbol{\omega_O}(\boldsymbol{\alpha}) \in \mathbb{R}^N$ is directly applied to the original prediction loss vector $\boldsymbol{l_O} \in \mathbb{R}^N$, resulting in the weighted average loss calculated as $mean(\boldsymbol{\omega_O}(\boldsymbol{\alpha}) \cdot \boldsymbol{l_O})$. Similarly, the weight vector $\boldsymbol{\omega_I}(\boldsymbol{\alpha}) \in \mathbb{R}^N$ not only reflects the overall significance of the mutual information but also each sample's individual contribution to this metric. The mean of these weights $mean(\boldsymbol{\omega_I}(\boldsymbol{\alpha}))$ represents the overall importance. For mutual information computations, $\boldsymbol{\omega_I}(\boldsymbol{\alpha})$ is transformed into a probability distribution $p_I^i = \frac{\omega_I^i}{\sum_{i=1}^{N} \omega_I^i}$. This adjustment affects the distribution used in mutual information calculations, necessitating a recalculation of mutual information as $I(\boldsymbol{\theta}, \boldsymbol{\beta}, \boldsymbol{\alpha})$, with details provided in Appendix A.1. As a result, the overall loss is formulated as:

$$\mathcal{L}(\boldsymbol{\theta}, \boldsymbol{\alpha}) = mean(\boldsymbol{\omega_O}(\boldsymbol{\alpha}) \cdot \boldsymbol{l_O}) + mean(\boldsymbol{\omega_I}(\boldsymbol{\alpha})) \cdot [-I(\boldsymbol{\theta}, \boldsymbol{\beta}, \boldsymbol{\omega_I}(\boldsymbol{\alpha}))] \tag{5}$$

### 3.2.2 Bi-level Optimization.

The ensuing challenge is optimizing the weighting network $\boldsymbol{\alpha}$. We achieve this by leveraging the supervision signals from a small validation dataset (Chen et al., 2022a). If the weighting network is properly optimized, the model trained with these weights is expected to show improved performance on the validation dataset in terms of the validation loss $\mathcal{L}_V(\boldsymbol{\theta}) = \frac{1}{M} \sum_j^M l_O^j(\boldsymbol{x}_j, \boldsymbol{y}_j)$, where $M$ denotes the size of the validation set. This constitutes a bi-level optimization problem (Hospedales et al., 2021; Chen et al., 2022a;b). At the inner level, model training is conducted through gradient descent:

$$\hat{\boldsymbol{\theta}}(\boldsymbol{\alpha}) = \boldsymbol{\theta} - \eta_1 \cdot \frac{\partial \mathcal{L}(\boldsymbol{\theta}, \boldsymbol{\alpha})}{\partial \boldsymbol{\theta}} \tag{6}$$

The objective is to ensure that the model performs optimally on the validation dataset:

$$\hat{\boldsymbol{\alpha}} = \boldsymbol{\alpha} - \eta_2 \cdot \frac{\partial \mathcal{L}_V(\boldsymbol{\theta}(\boldsymbol{\alpha}))}{\partial \boldsymbol{\alpha}} \tag{7}$$

Both $\eta_1$ and $\eta_2$ represent the learning rates for the respective optimization steps. Through the minimization of the validation loss, we aim to optimize the weighting network $\boldsymbol{\alpha}$.

## 4 Experimental Results

To demonstrate the versatility of our *LLM-TS Integrator*, we conduct extensive experiments across five main tasks: short- and long-term forecasting, imputation, classification, and anomaly detection. To maintain experimental integrity, our methodology adheres to the setup in (Wu et al., 2023). We detail the experimental setting in Appendix A.2.

### 4.0.1 Baselines.

Our evaluation employs a comprehensive array of baseline models across several architectural designs **(1)** CNN-based models, specifically TimesNet (Wu et al., 2023); **(2)** MLP-based models, including LightTS (Zhang et al., 2022) and DLinear (Zeng et al., 2023); **(3)** Transformer-based models, such as Reformer (Kitaev et al., 2020), Informer (Zhou et al., 2021b), Autoformer (Wu et al., 2021), FEDformer (Zhou et al., 2022), Nonstationary Transformer (Liu et al., 2022b), ETSformer (Woo et al., 2022), and PatchTST (Nie et al., 2023); **(4)** LLM-based models, represented by GPT4TS (Zhou et al., 2023). While we assess a wide range of models, we focus our discussion on the top-performing ones as highlighted in Zhou et al. (2023).

Additional comparisons for forecasting tasks include LLM-based models like Time-LLM (Jin et al., 2023) and TEST (Sun et al., 2023). For short-term forecasting, models like N-HiTS (Challu et al., 2023) and N-BEATS (Oreshkin et al., 2020) are included. Anomaly detection tasks are assessed using Anomaly Transformer (Xu et al., 2022), and for classification, models such as XGBoost (Chen and Guestrin, 2016), Rocket (Dempster et al., 2020), LSTNet (Lai et al., 2018), LSSL (Gu et al., 2022), Pyraformer (Liu et al., 2022a), TCN (Franceschi et al., 2019), and Flowformer (Huang et al., 2022) are considered. This broad selection of baselines enables a rigorous comparison across various tasks, highlighting the capabilities of our method.

### 4.1 Main Results

Figure 2 demonstrates that our *LLM-TS Integrator* consistently outperforms other methods in various tasks, underscoring its efficacy. We will refer to our method as *LLM-TS* in the tables for brevity. Unless otherwise indicated, we cite results from TimesNet (Wu et al., 2023). We reproduce TimesNet and GPT4TS (Zhou et al., 2023) experiments for all tasks. All results are averages from three runs with different seeds. Standard deviations for ablation studies are detailed in Appendix 8. The best results are highlighted in bold, with the second-best underlined. We also (1) present several showcases of our method in Appendix A.4 and (2) discuss the model efficiency in Appendix A.6

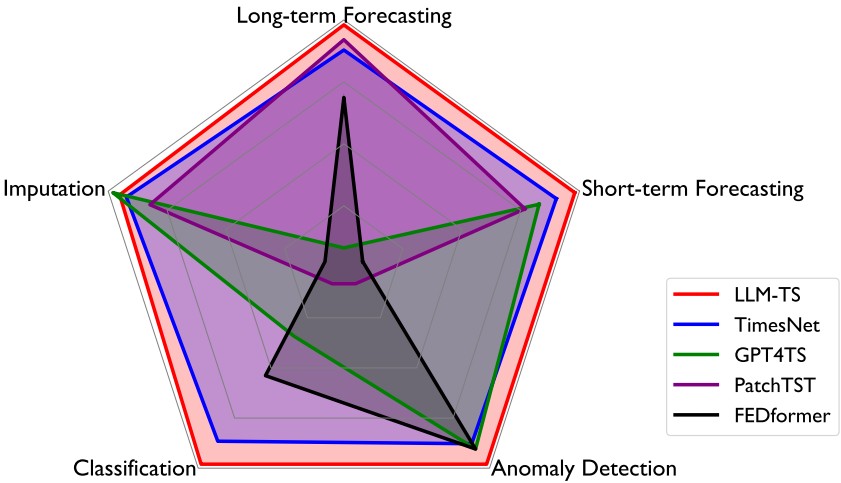

Figure 2: Model performance across different tasks.

## 4.2 Short- and Long- Term Forecasting

### 4.2.1 Setup.

To comprehensively assess our framework's forecasting capabilities, we engage it in both short- and long-term forecasting settings. In the realm of short-term forecasting, we utilize the M4 dataset (Spyros Makridakis, 2018), which aggregates univariate marketing data on a yearly, quarterly, and monthly basis. For long-term forecasting, we examine five datasets following (Zhou et al., 2023): ETT (Zhou et al., 2021a), Electricity (UCI, 2015), Traffic (PeMS, 2024), Weather (Wetterstation, 2024), and ILI (CDC, 2024). We adhere to the TimesNet setting with an input length of 96. For LLM-based methods like GPT4TS and Time-LLM, which use different input lengths, we rerun the experiments using their code. For PatchTST, we cite the results from (Wang et al., 2023a), as the original PatchTST uses an input length of 512. Due to shorter input lengths in this study compared to the original, the reported performance is lower.

### 4.2.2 Results.

As shown in Tables 1 and 2, our *LLM-TS* performs exceptionally well in both short- and long-term settings. It consistently surpasses TimesNet, highlighting the effectiveness of incorporating LLM-derived insights. Furthermore, it generally outperforms other LLM-based methods such as GPT4TS, TIME-LLM, and TEST, underscoring the advantages of integrating traditional TS modeling.

Table 1: Short-term M4 forecasting. The prediction lengths are in [6, 48] and results are obtained by weighting averages across multiple datasets with varying sampling intervals. Full results are in Appendix A.7.

| Methods | LLM-TS | TimesNet | GPT4TS | TIME-LLM | TEST | PatchTST | N-HiTS | N-BEATS | FEDformer | Stationary | Autoformer |
|---|---|---|---|---|---|---|---|---|---|---|---|
| SMAPE | **11.819** | 11.908 | 11.991 | 11.983 | 11.927 | 12.059 | 11.927 | 11.851 | 12.840 | 12.780 | 12.909 |
| MASE | **1.588** | 1.612 | 1.600 | 1.595 | 1.613 | 1.623 | 1.613 | 1.599 | 1.701 | 1.756 | 1.771 |
| OWA | **0.851** | 0.860 | 0.861 | 0.859 | 0.861 | 0.869 | 0.861 | 0.855 | 0.918 | 0.930 | 0.939 |

Table 2: Long-term forecasting: Averages over 4 lengths: 24, 36, 48, 60 for ILI, and 96, 192, 336, 720 for others. Full results in Appendix A.8.

| Methods | LLM-TS | | TimesNet | | TIME-LLM | | DLinear | | PatchTST | | GPT4TS | | FEDformer | | TEST | | Stationary | | ETSformer | |
|---|---|---|---|---|---|---|---|---|---|---|---|---|---|---|---|---|---|---|---|---|
| | MSE | MAE | MSE | MAE | MSE | MAE | MSE | MAE | MSE | MAE | MSE | MAE | MSE | MAE | MSE | MAE | MAE | MSE | MAE | MSE |
| Weather | **0.257** | **0.285** | 0.265 | 0.290 | 0.279 | 0.296 | 0.265 | 0.317 | 0.265 | **0.285** | 0.275 | 0.292 | 0.309 | 0.360 | 0.291 | 0.315 | 0.288 | 0.314 | 0.271 | 0.334 |
| ETTh1 | 0.454 | **0.451** | 0.470 | 0.462 | 0.474 | 0.459 | 0.456 | 0.452 | 0.516 | 0.484 | 0.473 | **0.451** | **0.440** | 0.460 | **0.440** | 0.460 | 0.570 | 0.537 | 0.542 | 0.510 |
| ETTh2 | 0.396 | 0.413 | 0.413 | 0.426 | 0.398 | 0.415 | 0.559 | 0.515 | 0.391 | **0.411** | **0.383** | 0.410 | 0.437 | 0.449 | 0.414 | 0.432 | 0.526 | 0.516 | 0.439 | 0.452 |
| ETTm1 | **0.401** | 0.409 | 0.414 | 0.418 | 0.437 | 0.421 | 0.403 | 0.407 | 0.406 | 0.407 | 0.408 | **0.400** | 0.448 | 0.452 | 0.402 | 0.411 | 0.481 | 0.456 | 0.429 | 0.425 |
| ETTm2 | 0.295 | **0.331** | 0.294 | **0.331** | 0.298 | 0.342 | 0.350 | 0.401 | **0.290** | 0.334 | 0.290 | 0.335 | 0.305 | 0.349 | 0.323 | 0.359 | 0.306 | 0.347 | 0.293 | 0.342 |
| ILI | **1.973** | **0.894** | 2.266 | 0.974 | 2.726 | 1.098 | 2.616 | 1.090 | 2.184 | 0.906 | 5.117 | 1.650 | 2.847 | 1.144 | 3.324 | 1.232 | 2.077 | 0.914 | 2.497 | 1.004 |
| ECL | 0.194 | 0.299 | 0.198 | 0.298 | 0.229 | 0.315 | 0.212 | 0.300 | 0.216 | 0.318 | 0.206 | **0.285** | 0.214 | 0.327 | 0.237 | 0.324 | **0.193** | 0.296 | 0.208 | 0.323 |
| Traffic | 0.618 | **0.333** | 0.627 | 0.335 | 0.606 | 0.395 | 0.625 | 0.383 | **0.529** | 0.341 | 0.561 | 0.373 | 0.610 | 0.376 | 0.581 | 0.388 | 0.624 | 0.340 | 0.621 | 0.396 |
| Average | **0.574** | **0.427** | 0.618 | 0.442 | 0.681 | 0.468 | 0.686 | 0.483 | 0.600 | 0.436 | 0.964 | 0.525 | 0.701 | 0.489 | 0.756 | 0.491 | 0.633 | 0.465 | 0.662 | 0.473 |

## 4.3 Imputation

### 4.3.1 Setup.

To assess our method's imputation capabilities, we employ three datasets: ETT (Zhou et al., 2021a), Electricity (UCI, 2015), and Weather (Wetterstation, 2024), serving as our benchmarks. To simulate various degrees of missing data, we randomly obscure time points at proportions of $\{12.5\%, 25\%, 37.5\%, 50\%\}$ following (Wu et al., 2023).

### 4.3.2 Results.

Table 3 illustrates that our method achieves performance comparable to GPT4TS and surpasses other baselines, highlighting its effectiveness. We attribute the robust performance of GPT4TS primarily to its backbone feature extractor: the pre-trained language model, which excels at capturing time series patterns, enhancing its imputation proficiency.

Table 3: Imputation task: Randomly masked $\{12.5\%, 25\%, 37.5\%, 50\%\}$ of points in 96-length series, averaging results over 4 mask ratios. Full results are in Appendix A.9.

| Methods | LLM-TS | | TimesNet | | GPT4TS | | PatchTST | | LightTS | | DLinear | | FEDformer | | Stationary | | Autoformer | | Reformer | |
|---|---|---|---|---|---|---|---|---|---|---|---|---|---|---|---|---|---|---|---|---|
| | MSE | MAE | MSE | MAE | MSE | MAE | MSE | MAE | MSE | MAE | MSE | MAE | MSE | MAE | MSE | MAE | MSE | MAE | MSE | MAE |
| ETTm1 | **0.025** | **0.103** | 0.028 | 0.109 | 0.028 | 0.108 | 0.047 | 0.140 | 0.104 | 0.218 | 0.093 | 0.206 | 0.062 | 0.177 | 0.036 | 0.126 | 0.051 | 0.150 | 0.055 | 0.166 |
| ETTm2 | **0.021** | **0.087** | 0.022 | 0.089 | 0.023 | 0.088 | 0.029 | 0.102 | 0.046 | 0.151 | 0.096 | 0.208 | 0.101 | 0.215 | 0.026 | 0.099 | 0.029 | 0.105 | 0.157 | 0.280 |
| ETTh1 | 0.087 | 0.198 | 0.090 | 0.199 | **0.069** | **0.174** | 0.115 | 0.224 | 0.284 | 0.373 | 0.201 | 0.306 | 0.117 | 0.246 | 0.094 | 0.201 | 0.103 | 0.214 | 0.122 | 0.245 |
| ETTh2 | **0.050** | 0.148 | 0.051 | 0.150 | **0.050** | **0.144** | 0.065 | 0.163 | 0.119 | 0.250 | 0.142 | 0.259 | 0.163 | 0.279 | 0.053 | 0.152 | 0.055 | 0.156 | 0.234 | 0.352 |
| ECL | 0.094 | 0.211 | 0.095 | 0.212 | 0.091 | 0.207 | **0.072** | **0.183** | 0.131 | 0.262 | 0.132 | 0.260 | 0.130 | 0.259 | 0.100 | 0.218 | 0.101 | 0.225 | 0.200 | 0.313 |
| Weather | **0.030** | 0.056 | 0.031 | 0.059 | 0.032 | 0.058 | 0.034 | **0.055** | 0.055 | 0.117 | 0.052 | 0.110 | 0.099 | 0.203 | 0.032 | 0.059 | 0.031 | 0.057 | 0.038 | 0.087 |
| Average | 0.051 | 0.134 | 0.053 | 0.136 | **0.049** | **0.130** | 0.060 | 0.144 | 0.123 | 0.228 | 0.119 | 0.224 | 0.112 | 0.229 | 0.056 | 0.142 | 0.061 | 0.151 | 0.134 | 0.240 |

## 4.4 Classification

### 4.4.1 Setup.

We focus on the application of our method to sequence-level time series classification tasks, a crucial test of its ability to learn high-level representations from data. Specifically, we employ 10 diverse multivariate datasets sourced from the UEA Time Series Classification repository (Bagnall et al., 2018). These datasets encompass a wide range of real-world applications, including gesture and action recognition, audio processing, among other practical domains. We reproduce the results of TEST based on their code (Sun et al., 2023).

### 4.4.2 Results.

As depicted in Figure 3, our *LLM-TS Integrator* achieves superior performance with an average accuracy of 73.4%. As detailed in Appendix A.10, it consistently outperforms other LLM-based methods across most

tasks, including GPT4TS and TEST. We attribute this enhanced capability to the traditional TS modeling techniques in our framework, which effectively capture classification characteristics more adeptly than LLMs.

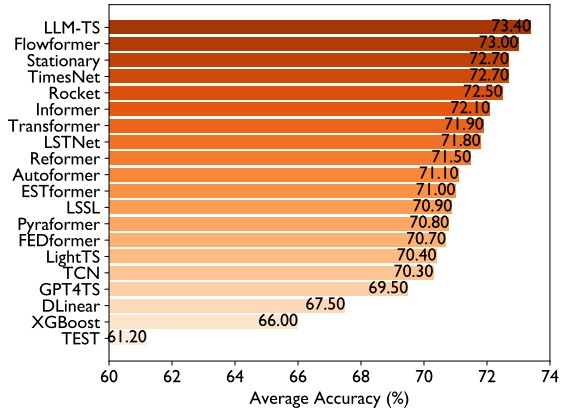

Figure 3: Model comparison in classification.

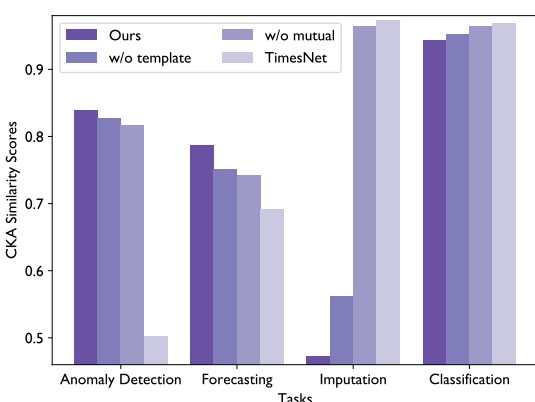

Figure 4: CKA by Task.

## 4.5 Anomaly Detection

### 4.5.1 Setup.

Our study concentrates on unsupervised time series anomaly detection, aiming to identify aberrant time points indicative of potential issues. We benchmark our method against five established anomaly detection datasets: SMD (Su et al., 2019), MSL (Hundman et al., 2018), SMAP (Hundman et al., 2018), SWaT (Mathur and Tippenhauer, 2016), and PSM (Abdulaal et al., 2021). These datasets span a variety of applications, including service monitoring, space and earth exploration, and water treatment processes. For a consistent evaluation framework across all experiments, we employ the classical reconstruction error metric to determine anomalies following Wu et al. (2023).

### 4.5.2 Results.

As indicated in Table 4, our *LLM-TS Integrator* exhibits superior performance with an average F1-score of 85.17%. This result underscores the versatility of *LLM-TS*, demonstrating its capability not only in classifying complete sequences, as discussed previously, but also in effectively detecting anomalies in time series data.

Table 4: Anomaly detection task. F1-score (as %). ∗. in the Transformers represents the name of ∗former. Full results are in Appendix A.11.

| Methods | LLM-TS | TimesNet | GPT4TS | PatchTS. | ETS. | FED. | LightTS | DLinear | Stationary | Auto. | Pyra. | Anomaly.[**] | In. | Re. | Trans. |
|---|---|---|---|---|---|---|---|---|---|---|---|---|---|---|---|
| SMD | 84.69 | 84.57 | 84.32 | 84.62 | 83.13 | 85.08 | 82.53 | 77.10 | 84.72 | 85.11 | 83.04 | 85.49 | 81.65 | 75.32 | 79.56 |
| MSL | 81.11 | 80.34 | 81.73 | 78.70 | 85.03 | 78.57 | 78.95 | 84.88 | 77.50 | 79.05 | 84.86 | 83.31 | 84.06 | 84.40 | 78.68 |
| SMAP | 69.41 | 69.18 | 68.86 | 68.82 | 69.50 | 70.76 | 69.21 | 69.26 | 71.09 | 71.12 | 71.09 | 71.18 | 69.92 | 70.40 | 69.70 |
| SWaT | 93.23 | 93.12 | 92.59 | 85.72 | 84.91 | 93.19 | 93.33 | 87.52 | 79.88 | 92.74 | 91.78 | 83.10 | 81.43 | 82.80 | 80.37 |
| PSM | 97.43 | 97.27 | 97.34 | 96.08 | 91.76 | 97.23 | 97.15 | 93.55 | 97.29 | 93.29 | 82.08 | 79.40 | 77.10 | 73.61 | 76.07 |
| Average | **85.17** | 84.90 | 84.97 | 82.79 | 82.87 | 84.97 | 84.23 | 82.46 | 82.08 | 84.26 | 82.57 | 80.50 | 78.83 | 77.31 | 76.88 |

## 4.6 Ablations

In this section, we first verify the effectiveness of our framework by sequentially removing key components: (1) *mutual information* module and (2) *sample reweighting* module. Additionally, for *mutual information*, we explore the impact of removing the template while retaining the raw time series data inputs to the LLM. We denote these variants as *w/o mutual*, *w/o reweight* and *w/o template*. Our experiments span long-term

Table 5: Results averaged over 4 prediction lengths.

| Methods | Ours | | w/o mutual | | w/o template | | w/o reweight | | TimesNet | |
|---|---|---|---|---|---|---|---|---|---|---|
| Metric | MSE | MAE | MSE | MAE | MSE | MAE | MSE | MAE | MSE | MAE |
| Weather | **0.257** | **0.285** | 0.264 | 0.290 | 0.263 | 0.288 | 0.264 | 0.291 | 0.265 | 0.290 |
| ETTh1 | **0.454** | **0.451** | 0.467 | 0.460 | 0.465 | 0.460 | 0.464 | 0.463 | 0.470 | 0.462 |
| ETTh2 | **0.396** | **0.413** | 0.411 | 0.423 | 0.404 | 0.418 | 0.408 | 0.419 | 0.413 | 0.426 |
| ETTm1 | **0.401** | **0.409** | 0.411 | 0.417 | 0.406 | 0.415 | 0.403 | 0.411 | 0.414 | 0.418 |
| ETTm2 | 0.295 | 0.331 | 0.300 | 0.335 | 0.298 | 0.332 | **0.292** | **0.328** | 0.294 | 0.331 |
| ILI | **1.973** | **0.894** | 2.221 | 0.942 | 2.173 | 0.950 | 2.173 | 0.947 | 2.266 | 0.974 |
| ECL | **0.194** | 0.299 | 0.199 | 0.302 | 0.196 | 0.301 | 0.199 | 0.299 | 0.198 | **0.298** |
| Traffic | **0.618** | **0.333** | 0.624 | 0.336 | 0.622 | 0.335 | 0.622 | 0.336 | 0.627 | 0.335 |

forecasting tasks including Weather, ETTh1, ETTm1 and ILI. As detailed in Table 5, the removal of any component leads to a decrease in performance, confirming the value of each design element. Additionally, we explore the use of the MINE estimator (Hjelm et al., 2019a) instead of the Jensen-Shannon MI estimator in our main paper, with further details provided in Appendix A.12. Lastly, we showcase various case studies to demonstrate the enhancements facilitated by our method in Appendix A.4 and explore template variations in Appendix A.5.

### 4.6.1 Mutual Information.

We further explore the *mutual information* module from a representation learning perspective, following the findings in Wu et al. (2023). They adopt a CKA (Centered Kernel Alignment) metric which measures similarity between representations obtained from the first and last layer of a model and they find that forecasting and anomaly detection benefits from high CKA similarity, contrasting with that imputation and classification tasks benefits from lower CKA similarity.

Experiments are conducted using the MSL dataset for the anomaly detection task, the Weather dataset for forecasting, the ETTh1 dataset for imputation, and the PEMS-SF dataset for classification. As depicted in Figure 4, the removal of components in our method results in decreased CKA similarity in anomaly detection and forecasting tasks, but an increase in imputation and classification tasks. This observation further substantiates the effectiveness of our components.

### 4.6.2 Sample Reweighting.

Regarding the *sample reweighting* module, we illustrate the behavior of the learned weighting network in Appendix A.12. The trend confirms our hypothesis: sample weight $\omega_O$ increases with the prediction loss $l_O$, and weight $\omega_I$ decreases as $l_O$ increases. This pattern validates our *sample reweighting* module. Further discussion comparing this module to a fixed weight scheme are presented in Appendix A.12.

To verify the effectiveness of our method, we conduct ablation studies focusing on (1) traditional time series (TS) models and (2) language models.

### 4.6.3 Traditional Models.

Although we utilize TimesNet as our primary model, our framework is applicable to other traditional models. We explored additional traditional models including ETSformer (Woo et al., 2022), Stationary (Liu et al., 2022b), and FreTS (Yi et al., 2023). As shown in Table 6, integrating *LLM-TS* generally enhances performance across all traditional models, underscoring the benefits of our method.

### 4.6.4 Language Models.

In the main paper, the LLaMA-3b model (Touvron et al., 2023) is used to generate embeddings for the TS language description. We compare it with GPT2 (Radford et al., 2019) and BERT (Devlin et al., 2019) to assess different embeddings' performance. Table 7 reveals that LLaMA-3b generally outperforms the

alternatives, and all LMs improve results compared to non-LLM approaches, validating the effectiveness of *LLM-TS Integrator*.

Table 6: Ablation results on different traditional models. Full results are in Appendix A.12.

| Methods | ETSformer | | ETS LLM-TS | | Stationary | | Stat LLM-TS | | FreTS | | FreTS LLM-TS | |
|---|---|---|---|---|---|---|---|---|---|---|---|---|
| Metric | MSE | MAE | MSE | MAE | MSE | MAE | MSE | MAE | MSE | MAE | MSE | MAE |
| *Weather* | 0.313 | 0.382 | 0.307 | 0.375 | 0.282 | 0.307 | 0.284 | 0.309 | 0.262 | 0.306 | 0.255 | 0.302 |
| *ETTh*1 | 0.799 | 0.684 | 0.791 | 0.678 | 0.667 | 0.582 | 0.653 | 0.572 | 0.484 | 0.473 | 0.478 | 0.466 |
| *ETTm*1 | 0.638 | 0.583 | 0.555 | 0.528 | 0.527 | 0.477 | 0.522 | 0.471 | 0.415 | 0.422 | 0.407 | 0.415 |
| *ILI* | 3.922 | 1.367 | 3.740 | 1.320 | 2.722 | 1.041 | 2.205 | 0.935 | 3.449 | 1.279 | 3.158 | 1.211 |

Table 7: Ablation results on different LLM embeddings. Full results are in Appendix A.12.

| Methods | LLM-TS (LLaMA) | | LLaMA w/o template | | GPT2 | | BERT | | No LLM | |
|---|---|---|---|---|---|---|---|---|---|---|
| Metric | MSE | MAE | MSE | MAE | MSE | MAE | MSE | MAE | MSE | MAE |
| *Weather* | **0.257** | **0.285** | 0.263 | 0.288 | 0.261 | 0.287 | 0.260 | 0.287 | 0.264 | 0.290 |
| *ETTh*1 | **0.454** | **0.451** | 0.465 | 0.460 | 0.464 | 0.458 | 0.467 | 0.460 | 0.467 | 0.460 |
| *ETTm*1 | **0.401** | **0.409** | 0.406 | 0.415 | 0.406 | 0.413 | 0.406 | 0.412 | 0.411 | 0.417 |
| *ILI* | **1.973** | **0.894** | 2.173 | 0.950 | 2.169 | 0.936 | 2.193 | 0.952 | 2.221 | 0.942 |

## 5 Related Work

### 5.1 LLM for TS Modeling.

FPT (Zhou et al., 2023) suggests utilizing pre-trained language models to extract features from time series for improved predictions. TIME-LLM (Jin et al., 2024) and TEST (Sun et al., 2023) adapt LLMs for general time series forecasting by maintaining the original language model structure while reprogramming the input to fit time series data (Zhou et al., 2024). LLMTIME (Gruver et al., 2023b) interprets time series as sequences of numbers, treating forecasting as a next-token prediction task akin to text processing, applying pre-trained LLMs for this purpose. Given that it is not a state-of-the-art method and primarily targets zero-shot forecasting, it has not been incorporated into our experimental framework. TEMPO (Cao et al., 2024) utilizes essential inductive biases of the TS task for generative pre-trained transformer models. Gao et al. (2024); Goswami et al. (2024) also explore time series foundation models by pre-training large models on extensive time series datasets.

### 5.2 Time Series to Text.

PromptCast (Xue and Salim, 2023) proposes to transform the numerical input and output into prompts, which enables forecasting in a sentence-to-sentence manner. Time-LLM (Jin et al., 2024) incorporates background, instruction and statistical information of the time series data via natural language to facilitate time series forecasting in LLM. LLMTIME (Gruver et al., 2023a) converts time series data into a string of numbers and predicts future values as if completing a text. AutoTimes (Liu et al., 2024) maps time series into the embedding space of language tokens, enabling autoregressive generation of future predictions.

### 5.3 Mutual Information

The Infomax principle (Linsker, 1988; Bell and Sejnowski, 1995), applied in the context of neural networks, advocates for maximizing mutual information between the inputs and outputs of a network. Traditionally, quantifying mutual information was challenging outside a few specific probability distributions, as discussed in Shwartz-Ziv and Tishby (2017). This complexity led to the development of various heuristics and approximations (Tishby et al., 2000). Recently, a breakthrough came with MINE (Belghazi et al., 2021), which introduced a neural estimator capable of assessing mutual information between two arbitrary quantities

with a precision that depends on the capacity of the encoding network. This innovative approach has spearheaded advancements in the field of representation learning (Hjelm et al., 2019b; Sylvain et al., 2020). The estimator we utilize is based on the Jensen-Shannon divergence variant of the MINE mutual information estimator.

### 5.4 Sample Reweighting.

Sample reweighting is commonly used to improve training efficacy (Fang et al., 2023; Wang et al., 2023b; Yuan et al., 2023a; Zhang et al., 2023; Yuan et al., 2023b). Traditional approaches (Freund and Schapire, 1997; Sun et al., 2007) assign larger weights to samples with higher loss values, as these hard samples have greater learning potential. Recent studies (Ren et al., 2018) suggest using a validation set to guide the learning of sample weights, which can enhance model training. Notably, meta-weight-net (Shu et al., 2019) proposes learning a mapping from sample loss to sample weight. In this work, we adopt an MLP network that takes sample prediction loss as input and outputs dual weights for prediction loss and mutual information loss.

## 6   Conclusion and Discussion

In conclusion, the *LLM-TS Integrator framework* offers a promising approach to integrating Large Language Models (LLMs) with traditional TS methods. By encouraging high mutual information between textual and TS data, our method aims to maintain the distinct characteristics of TS while benefiting from the advanced pattern recognition capabilities of LLMs. The introduced sample reweighting module enhances performance by dynamically adjusting the relevance of each sample based on its predictive and informational contributions. Comprehensive empirical evaluations suggest that this framework improves accuracy across various TS tasks, including forecasting, anomaly detection, and classification. However, further research is necessary to confirm these results across more diverse and robust datasets including Monash (Godahewa et al., 2021) and GIFT-Eval (Aksu et al., 2024), and to explore additional LLM features that could further enrich our model. Additionally, it is important to acknowledge current limitations, such as the need for computational resources and the potential challenges in aligning the two modalities effectively.

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

# A  Appendix

## A.1  Mutual Information Recalculation

Recall that mutual information can be calculated using the equation:

$$I(\boldsymbol{\theta}, \boldsymbol{\beta}) = \mathbb{E}_{\mathbb{S}}[-sp(-T_{\boldsymbol{\beta}}(\boldsymbol{h}_{\boldsymbol{\theta}}^m(\boldsymbol{x}), \boldsymbol{h}^l(\boldsymbol{t})))] - \mathbb{E}_{\mathbb{S} \times \tilde{\mathbb{S}}}[sp(T_{\boldsymbol{\beta}}(\boldsymbol{h}_{\boldsymbol{\theta}}^m(\boldsymbol{x}), \boldsymbol{h}^l(\tilde{\boldsymbol{t}})))], \tag{8}$$

where $T_{\boldsymbol{\beta}}$ signifies the discriminator characterized by parameters $\boldsymbol{\beta}$, and sp denotes the softplus function. Notably, $(\boldsymbol{x}, \boldsymbol{t})$ symbolizes a sample from the dataset $\mathbb{S}$, while $(\tilde{\boldsymbol{x}}, \tilde{\boldsymbol{t}})$ represents a different sample from the dataset $\tilde{\mathbb{S}} = \mathbb{S}$.

This formulation presumes a uniform distribution of samples. However, we have already computed probabilities $p_I^i$ for each sample, which introduces a non-uniform distribution. For a batch of $N$ samples, the expected value is computed as

$$\mathbb{E}_{\mathbb{S}}[-sp(-T_{\boldsymbol{\beta}}(\boldsymbol{h}_{\boldsymbol{\theta}}^m(\boldsymbol{x}), \boldsymbol{h}^l(\boldsymbol{t})))] = -\sum_{i=1}^{N} p_I^i\, sp(-T_{\boldsymbol{\beta}}(\boldsymbol{h}_{\boldsymbol{\theta}}^m(\boldsymbol{x^i}), \boldsymbol{h}^l(\boldsymbol{t^i}))). \tag{9}$$

$$\mathbb{E}_{\mathbb{S} \times \tilde{\mathbb{S}}}[sp(T_{\boldsymbol{\beta}}(\boldsymbol{h}_{\boldsymbol{\theta}}^m(\boldsymbol{x}), \boldsymbol{h}^l(\tilde{\boldsymbol{t}})))] = \sum_{i} \sum_{i \neq j} \hat{p}^{ij}\, sp(T_{\boldsymbol{\beta}}(\boldsymbol{h}_{\boldsymbol{\theta}}^m(\boldsymbol{x^i}), \boldsymbol{h}^l(\tilde{\boldsymbol{t^j}}))). \tag{10}$$

Here, $\hat{p}^{ij}$ is defined as $\frac{p_I^i \cdot p_I^j}{\sum_i \sum_{i \neq j} p_I^i \cdot p_I^j}$, adjusting for the non-uniform distribution of sample probabilities. As $p_I^i$ is produced from the weighting network $\boldsymbol{\alpha}$, we can also write $I(\boldsymbol{\theta}, \boldsymbol{\beta})$ as $I(\boldsymbol{\theta}, \boldsymbol{\beta}, \boldsymbol{\alpha})$.

## A.2  Experimental Settings

Following Shu et al. (2019), the weighting network comprises a two-layer MLP with a hidden size of 100, and we set the learning rate $\eta_2$ for this network at 0.001. The learning rate $\eta_0$ of the discriminator is set as 0.001 at the first epoch and then decreases to 0.0001 for the rest of epochs.

## A.3  Standard Error Results

Table 8 presents the results along with standard errors to underscore the consistency and reliability of our method's performance.

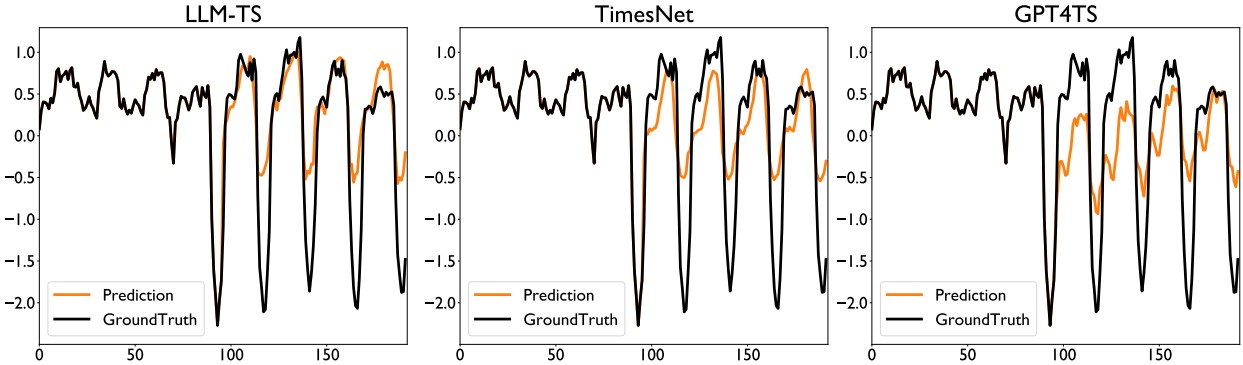

Figure 5: ETTh1

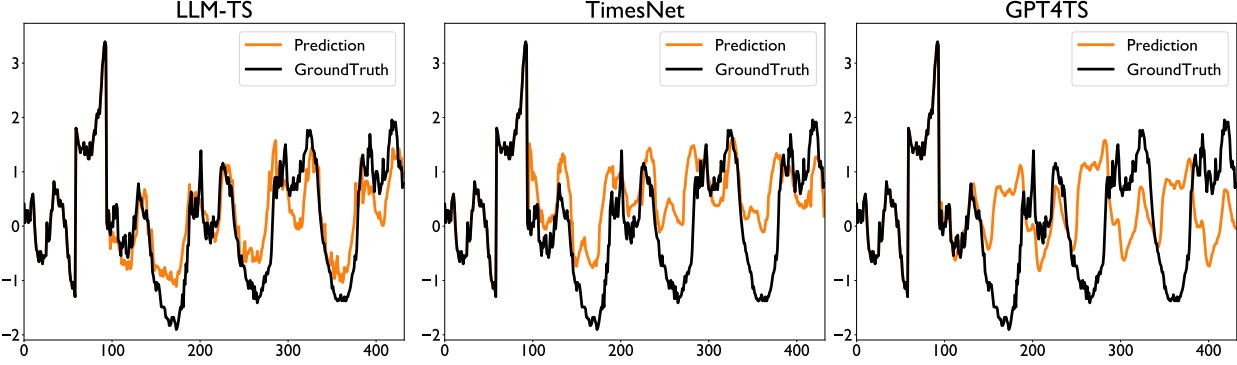

Figure 6: ETTm1

Table 8: Ablation results with standard error.

| Methods | | Ours | | w/o mutual | | w/o reweight | | TimesNet | |
|---|---|---|---|---|---|---|---|---|---|
| Metric | | MSE | MAE | MSE | MAE | MSE | MAE | MSE | MAE |
| *Weather* | 96 | $0.166 \pm 0.001$ | $0.217 \pm 0.001$ | $0.168 \pm 0.002$ | $0.218 \pm 0.003$ | $0.181 \pm 0.002$ | $0.232 \pm 0.001$ | $0.174 \pm 0.002$ | $0.224 \pm 0.001$ |
| | 192 | $0.229 \pm 0.002$ | $0.269 \pm 0.002$ | $0.227 \pm 0.002$ | $0.268 \pm 0.002$ | $0.230 \pm 0.001$ | $0.270 \pm 0.002$ | $0.235 \pm 0.001$ | $0.272 \pm 0.002$ |
| | 336 | $0.278 \pm 0.001$ | $0.302 \pm 0.002$ | $0.298 \pm 0.002$ | $0.318 \pm 0.002$ | $0.283 \pm 0.002$ | $0.306 \pm 0.001$ | $0.285 \pm 0.002$ | $0.307 \pm 0.001$ |
| | 720 | $0.354 \pm 0.001$ | $0.351 \pm 0.001$ | $0.361 \pm 0.001$ | $0.356 \pm 0.001$ | $0.361 \pm 0.001$ | $0.355 \pm 0.001$ | $0.365 \pm 0.001$ | $0.358 \pm 0.000$ |
| *ETTh1* | 96 | $0.403 \pm 0.003$ | $0.420 \pm 0.002$ | $0.402 \pm 0.002$ | $0.422 \pm 0.001$ | $0.408 \pm 0.002$ | $0.428 \pm 0.001$ | $0.414 \pm 0.003$ | $0.431 \pm 0.002$ |
| | 192 | $0.440 \pm 0.005$ | $0.441 \pm 0.002$ | $0.459 \pm 0.003$ | $0.455 \pm 0.003$ | $0.469 \pm 0.003$ | $0.460 \pm 0.002$ | $0.463 \pm 0.006$ | $0.456 \pm 0.003$ |
| | 336 | $0.471 \pm 0.003$ | $0.457 \pm 0.002$ | $0.471 \pm 0.003$ | $0.457 \pm 0.002$ | $0.492 \pm 0.002$ | $0.474 \pm 0.002$ | $0.487 \pm 0.004$ | $0.466 \pm 0.003$ |
| | 720 | $0.503 \pm 0.003$ | $0.487 \pm 0.002$ | $0.535 \pm 0.002$ | $0.507 \pm 0.002$ | $0.485 \pm 0.003$ | $0.478 \pm 0.004$ | $0.517 \pm 0.002$ | $0.494 \pm 0.002$ |
| *ETTm1* | 96 | $0.329 \pm 0.008$ | $0.371 \pm 0.004$ | $0.341 \pm 0.006$ | $0.377 \pm 0.005$ | $0.350 \pm 0.006$ | $0.387 \pm 0.003$ | $0.340 \pm 0.006$ | $0.377 \pm 0.004$ |
| | 192 | $0.380 \pm 0.005$ | $0.398 \pm 0.002$ | $0.404 \pm 0.006$ | $0.413 \pm 0.003$ | $0.383 \pm 0.006$ | $0.397 \pm 0.003$ | $0.406 \pm 0.007$ | $0.408 \pm 0.002$ |
| | 336 | $0.418 \pm 0.002$ | $0.425 \pm 0.002$ | $0.432 \pm 0.003$ | $0.428 \pm 0.001$ | $0.410 \pm 0.002$ | $0.411 \pm 0.002$ | $0.424 \pm 0.003$ | $0.425 \pm 0.002$ |
| | 720 | $0.476 \pm 0.005$ | $0.440 \pm 0.003$ | $0.468 \pm 0.005$ | $0.449 \pm 0.002$ | $0.467 \pm 0.004$ | $0.448 \pm 0.002$ | $0.485 \pm 0.006$ | $0.461 \pm 0.004$ |
| *ILI* | 24 | $1.921 \pm 0.116$ | $0.898 \pm 0.019$ | $2.170 \pm 0.100$ | $0.947 \pm 0.023$ | $1.934 \pm 0.098$ | $0.925 \pm 0.020$ | $2.072 \pm 0.122$ | $0.948 \pm 0.015$ |
| | 36 | $2.151 \pm 0.035$ | $0.933 \pm 0.020$ | $2.093 \pm 0.069$ | $0.889 \pm 0.024$ | $2.505 \pm 0.103$ | $1.020 \pm 0.010$ | $2.494 \pm 0.072$ | $1.019 \pm 0.004$ |
| | 48 | $2.062 \pm 0.052$ | $0.892 \pm 0.011$ | $2.418 \pm 0.034$ | $0.959 \pm 0.008$ | $2.325 \pm 0.116$ | $0.948 \pm 0.036$ | $2.298 \pm 0.038$ | $0.964 \pm 0.006$ |
| | 60 | $1.759 \pm 0.124$ | $0.853 \pm 0.035$ | $2.203 \pm 0.105$ | $0.971 \pm 0.028$ | $1.926 \pm 0.088$ | $0.896 \pm 0.023$ | $2.198 \pm 0.040$ | $0.963 \pm 0.010$ |

## A.4    Showcases

To provide a clear comparison among different models, we showcase the forecasting task results on ETTh1 (96-96) and ETTm1 (96-336) using three models: *LLM-TS*, TimesNet, and GPT4TS. As shown in Figures 5 and 6, our LLM-TS model produces significantly more accurate predictions, demonstrating its effectiveness.

To illustrate the performance improvements achieved by the LLM-TS Integrator framework, we introduce a case study. We created a training set with a weighted sine function:

$$\sum_{i=1}^{4} \omega_i \sin(f_i t + p_i) + \epsilon N(0, 1) \tag{11}$$

where $w_1 = 0.1$, $w_2 = 0.2$, $w_3 = 0.3$, $w_4 = 0.4$; $f_1 = \frac{1}{40}$, $f_2 = \frac{1}{45}$, $f_3 = \frac{1}{50}$, $f_4 = \frac{1}{55}$; $p_1 = 0$, $p_2 = 1$, $p_3 = 2$, $p_4 = 3$; and $\epsilon = 0.1$ is the noise level. We generated a long sequence of length $10,000$ and then sampled a batch of size 64 with a sequence length of 96 and a prediction length of 336 to train GPT4TS, TimesNet, and LLM-TS on this data for $1,000$ iterations. For testing, we created a test set with frequency $f = \frac{1}{20}$, which is greater than $\max(f_1, f_2, f_3, f_4)$, and used $p = 2.5$, $w = 1$ and $\epsilon = 0.1$.

As shown in Figure 7, Figure 8 an Figure 9, we can know:

- GPT4TS fails to accurately capture periodic information as it relies solely on a language model without incorporating traditional mathematical modelling.

- TimesNet generally captures periodic information due to the use of the FFT mathematical operator, but it still does not perfectly match the ground truth.

- LLM-TS captures periodic information and better matches the ground truth by integrating rich language model insights into the traditional TimesNet model.

This case study highlights how the LLM-TS Integrator framework benefits from both inherent properties of traditional TS models and pattern recognition abilities of LLMs, demonstrating the effectiveness of our approach.

## A.5    Template Variation

We conducted additional experiments on the ETTh1 dataset for long-term forecasting with GPT2. The original template achieves a Mean Squared Error (MSE) of 0.464 and a Mean Absolute Error (MAE) of 0.458.

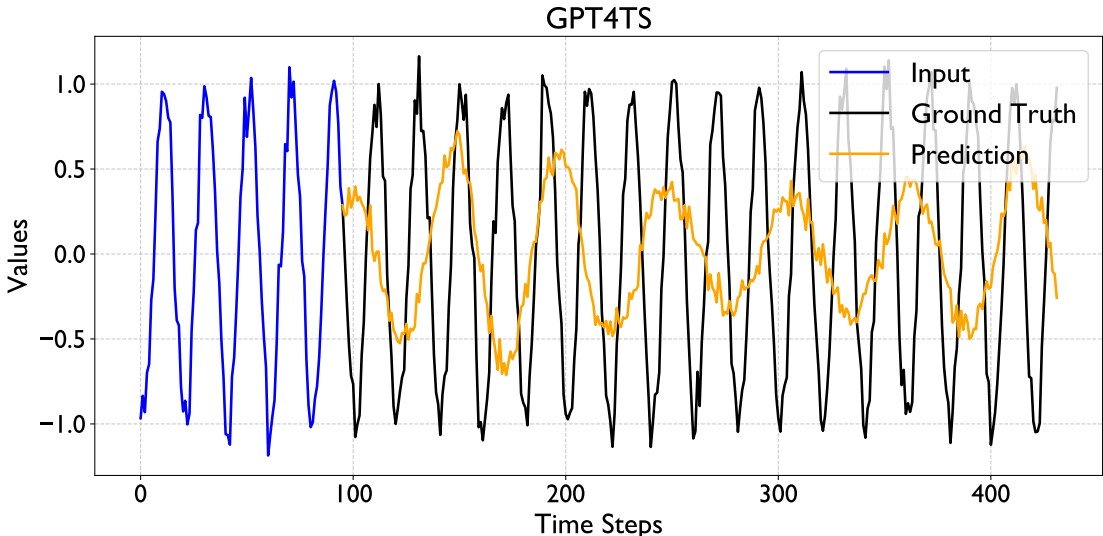

Figure 7: GPT4TS on synthetic data

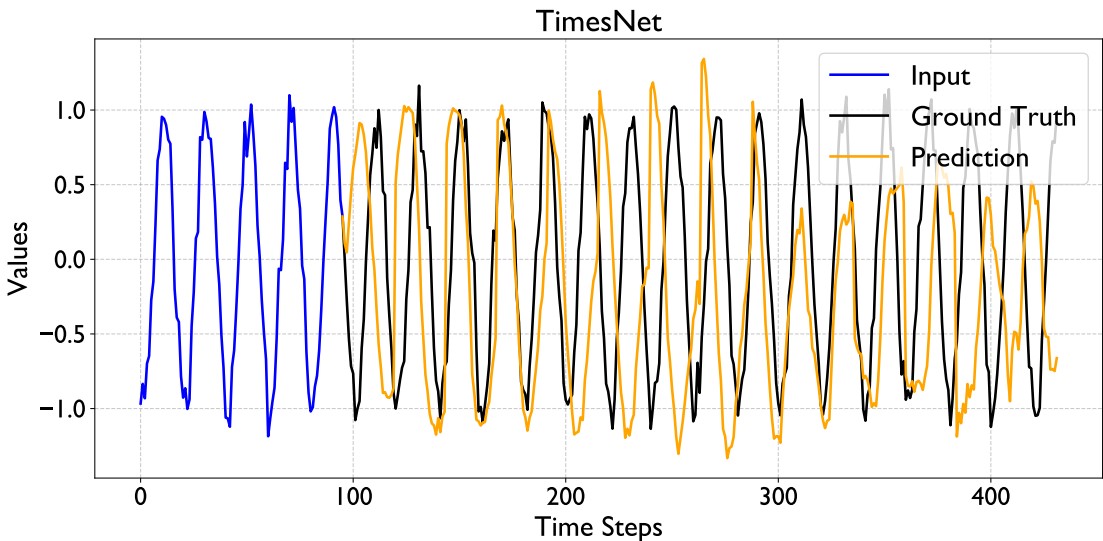

Figure 8: TimesNet on synthetic data

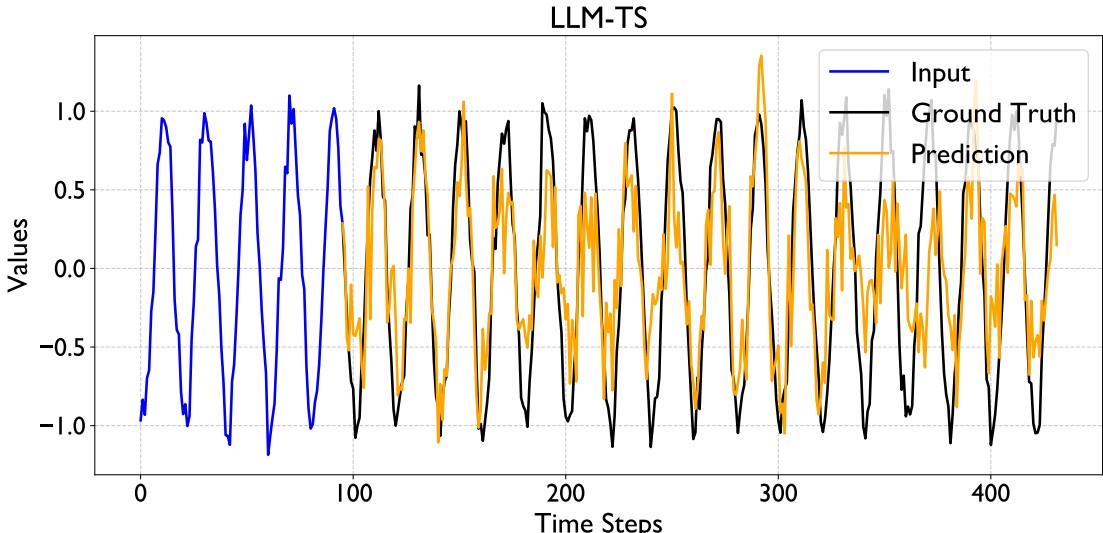

Figure 9: LLM-TS on synthetic data

We tested variations of the template by changing the original context from "The Electricity Transformer Temperature is a crucial indicator in electric power long-term deployment." to:

- Variation 1: "The temperature of the electricity transformer is a vital metric for long-term electric power deployment."

- Variation 2: "Monitoring the temperature of electricity transformers is essential for the long-term deployment of electric power."

- Variation 3: "The temperature of electricity transformers serves as a key indicator in the long-term deployment of electric power."

- Variation 4: No template.

Besides, we also consider the following changes:

- w/o Input Statistics: excluding input statistical data from our analysis.

- w/o Mean, Max, Median: remove mean, max and median informatino.

- w/o Lags: remove lags information.

The performance of these variations is summarized in Table 9. These results indicate that the performance is quite similar across different variations, supporting the robustness of our approach regardless of minor template modifications. For further details on the template implementation, refer to our code repository at `https://anonymous.4open.science/r/llm_ts_anonymous-F07D/utils/tools.py`.

### A.6   Model Efficiency Analysis

Compared to TimesNet, our *LLM-TS integrator* introduces additional costs due to the mutual information and sample weighting modules. However, after training, the inference cost of our method is the same as TimesNet. We detail the time cost of each component for ETTh1 and ETTm1 tasks, using a batch size of 32 on a 32G V100 GPU. As shown in Table 10, the training cost of our method is reasonable, given that it achieves the best performance across most tasks.

Table 9: Performance across different template variations (standard error)

| Template Variation | MSE | MAE |
|---|---|---|
| Original Template | $0.464 \pm 0.002$ | $0.458 \pm 0.003$ |
| Variation 1 | $0.460 \pm 0.003$ | $0.456 \pm 0.002$ |
| Variation 2 | $0.465 \pm 0.003$ | $0.460 \pm 0.003$ |
| Variation 3 | $0.464 \pm 0.002$ | $0.459 \pm 0.002$ |
| Variation 4 (No template) | $0.466 \pm 0.003$ | $0.460 \pm 0.003$ |
| w/o Input Statistics | $0.468 \pm 0.002$ | $0.462 \pm 0.002$ |
| w/o Mean/Max/Median | $0.465 \pm 0.002$ | $0.459 \pm 0.002$ |
| w/o Lags | $0.467 \pm 0.002$ | $0.460 \pm 0.003$ |

It is important to note that we use the pre-trained LLM to obtain the text embeddings only once. These embeddings can then be used throughout the training process. For instance, obtaining the embeddings for the ETTh1 dataset using the llama-3b model on an A100 GPU takes approximately 1 hour. We acknowledge that embedding extraction may become more time-consuming and memory-intensive with significantly larger or multivariate datasets. In such scenarios, strategies such as prioritizing difficult or representative data points could be employed to manage computational costs effectively. After this, the embeddings are utilized in our framework to train the model, and in the final output of the TimesNet model. This ensures that the inference time of our method is identical to that of the TimesNet model.

As detailed in the TimesNet paper, our backbone model TimesNet is relatively small with 0.067 MB parameters. For comparison, other models have the following sizes: Non-stationary Transformer has 1.884 MB, Autoformer has 1.848 MB, FEDformer has 2.9 MB, LightTS has 0.163 MB, DLinear has 0.296 MB, ETSformer has 1.123 MB, Informer has 1.903 MB, Reformer has 1.157 MB, and Pyraformer has 1.308 MB. The introduced mutual information network consists of only two linear layers of size $64x64$ and $64x4096$, which is negligible in terms of additional parameters. Similarly, the introduced MLP network consists of four layers: $1 \times 100$, $100 \times 1$, $1 \times 1$, and $1 \times 1$, and the number of parameters is also negligible.

Thus, our model remains very small and efficient, with inference time identical to TimesNet (as the mutual information component is only used during training). Given that many TS models are primarily used for inference, our approach offers effective performance gains with minimal additional computational cost.

From a theoretical standpoint, let $d_\alpha$ denote the dimension of the weighting network parameters and $d_\theta$ denote the dimension of the prediction network parameters. Denote the computation cost of Eq. (6) as $c(d_\alpha, d_\theta)$. According to Section 4 of Franceschi et al. (2017), the time complexity of the bi-level optimization is $O(c(d_\alpha, d_\theta))$, with the primary computational cost arising from the Jacobian-vector product in Eq. (7). This explains the time complexity of the sample reweighting module.

Table 10: Cost Comparison per step(s).

| Methods | Overall | TimesNet | Mutual Information | Sample Reweighting |
|---|---|---|---|---|
| ETTh1 | 3.177 | 0.126 | 0.577 | 2.474 |
| Weather | 5.563 | 0.436 | 1.094 | 4.033 |

## A.7 Full Results of Short-term Forecasting

Table 11 displays the comprehensive results for short-term forecasting.

## A.8 Full Results of Long-Term Forecasting

Full results for long-term forecasting are presented in Table 12.

Table 11: Full results of short-term forecasting.

| | Methods | LLM-TS | TimesNet | GPT4TS | TIME-LLM | PatchTST | N-HiTS | N-BEATS | FEDformer | Stationary | Autoformer |
|---|---|---|---|---|---|---|---|---|---|---|---|
| *Yearly* | *SMAPE* | 13.369 | 13.512 | 13.531 | 13.419 | 13.477 | 13.418 | 13.436 | 13.728 | 13.717 | 13.974 |
| | *MASE* | 3.021 | 3.065 | 3.015 | 3.0050 | 3.019 | 3.045 | 3.043 | 3.048 | 3.078 | 3.134 |
| | *OWA* | 0.789 | 0.799 | 0.793 | 0.789 | 0.792 | 0.793 | 0.794 | 0.803 | 0.807 | 0.822 |
| *Quarterly* | *SMAPE* | 10.020 | 10.069 | 10.177 | 10.110 | 10.38 | 10.202 | 10.124 | 10.792 | 10.958 | 11.338 |
| | *MASE* | 1.162 | 1.178 | 1.194 | 1.178 | 1.233 | 1.194 | 1.169 | 1.283 | 1.325 | 1.365 |
| | *OWA* | 0.878 | 0.887 | 0.898 | 0.889 | 0.921 | 0.899 | 0.886 | 0.958 | 0.981 | 1.012 |
| *Monthly* | *SMAPE* | 12.696 | 12.783 | 12.894 | 12.980 | 12.959 | 12.791 | 12.677 | 14.260 | 13.917 | 13.958 |
| | *MASE* | 0.936 | 0.949 | 0.956 | 0.963 | 0.970 | 0.969 | 0.937 | 1.102 | 1.097 | 1.103 |
| | *OWA* | 0.880 | 0.889 | 0.897 | 0.903 | 0.905 | 0.899 | 0.880 | 1.012 | 0.998 | 1.002 |
| *Others* | *SMAPE* | 4.916 | 4.954 | 4.940 | 4.795 | 4.952 | 5.061 | 4.925 | 4.954 | 6.302 | 5.485 |
| | *MASE* | 3.310 | 3.364 | 3.228 | 3.178 | 3.347 | 3.216 | 3.391 | 3.264 | 4.064 | 3.865 |
| | *OWA* | 1.039 | 1.052 | 1.029 | 1.006 | 1.049 | 1.040 | 1.053 | 1.036 | 1.304 | 1.187 |
| *Average* | *SMAPE* | 11.819 | 11.908 | 11.991 | 11.983 | 12.059 | 11.927 | 11.851 | 12.840 | 12.780 | 12.909 |
| | *MASE* | 1.588 | 1.612 | 1.600 | 1.595 | 1.623 | 1.613 | 1.599 | 1.701 | 1.756 | 1.771 |
| | *OWA* | 0.851 | 0.860 | 0.861 | 0.859 | 0.869 | 0.861 | 0.855 | 0.918 | 0.930 | 0.939 |

### A.9 Full Results of Imputation.

Table 13 contains the detailed results of our imputation tasks.

### A.10 Full Results of Classification

Table 14 contains the comprehensive results for classification.

### A.11 Full Results of Anamoly Detection

Full results for anamoly detection are detailed in Table 15.

### A.12 Further Ablation Studies

**MLP.** Given that our template comprises only simple summary statistics of the time series and the original values, we investigated whether the LLM component is essential. To this end, we replaced the LLM with an MLP that accepts the normalized concatenated statistics (mean, max, median, lags) and outputs an embedding of identical dimension to the LLM. We conducted experiments on ETTh1 and ETTm1. For ETTm1, the MSE increased from 0.401 to 0.420, and the MAE increased from 0.409 to 0.425. Similarly, for ETTh1, the MSE increased from 0.454 to 0.478, and the MAE increased from 0.451 to 0.463. We also attempted using a single-layer self-attention module instead of the MLP. However, the performance remained suboptimal. On ETTm1, the MSE increased from 0.401 to 0.423 and the MAE from 0.409 to 0.419; similarly, on ETTh1, the MSE increased from 0.454 to 0.475 and the MAE from 0.451 to 0.461. These results demonstrate that the LLM component is crucial in effectively capturing the mutual information from the template, thereby justifying its use in our approach. Tan et al. (2024) demonstrate that using LLMs as prediction backbones may not yield substantial improvements; however, our results highlight the effectiveness of LLM embeddings when integrated as auxiliary information within traditional time-series models.

**Mutual Information Estimator.** In the main paper, we utilize the Jensen-Shannon mutual information (MI) estimator. Additionally, we explore the Mutual Information Neural Estimator (MINE) (Hjelm et al., 2019a). We evaluate both estimators on two tasks, ETTh1 and ETTm1, with results averaged over four prediction lengths. For ETTh1, the MSE and MAE using the original Jensen-Shannon estimator are 0.454 and 0.451, respectively, compared to 0.460 and 0.457 with MINE. For ETTm1, the MSE and MAE are 0.401 and 0.409 with the original estimator, and 0.402 and 0.410 with MINE. These comparisons highlight the robustness of our method across different mutual information estimators.

Table 12: Full results for long-term forecasting. We use prediction length $O \in \{96, 192, 336, 720\}$ except for ILI and $O \in \{24, 36, 48, 60\}$ for ILI. A lower MSE indicates better performance.

| Methods | | LLM-TS | | TimesNet | | TIME-LLM | | DLinear | | PatchTST | | GPT4TS | | FEDformer | | TEST | | Stationary | | ETSformer | |
|---|---|---|---|---|---|---|---|---|---|---|---|---|---|---|---|---|---|---|---|---|---|
| Metric | | MSE | MAE | MSE | MAE | MSE | MAE | MSE | MAE | MSE | MAE | MSE | MAE | MSE | MAE | MSE | MAE | MSE | MAE | MSE | MAE |
| Weather | 96 | 0.166 | 0.217 | 0.174 | 0.224 | 0.202 | 0.239 | 0.196 | 0.255 | 0.186 | 0.227 | 0.196 | 0.234 | 0.217 | 0.296 | 0.214 | 0.264 | 0.173 | 0.223 | 0.197 | 0.281 |
| | 192 | 0.229 | 0.269 | 0.235 | 0.272 | 0.245 | 0.277 | 0.237 | 0.296 | 0.234 | 0.265 | 0.241 | 0.271 | 0.276 | 0.336 | 0.262 | 0.298 | 0.245 | 0.285 | 0.237 | 0.312 |
| | 336 | 0.278 | 0.302 | 0.235 | 0.272 | 0.300 | 0.313 | 0.283 | 0.335 | 0.284 | 0.301 | 0.296 | 0.308 | 0.339 | 0.380 | 0.310 | 0.329 | 0.321 | 0.338 | 0.298 | 0.353 |
| | 720 | 0.354 | 0.351 | 0.365 | 0.358 | 0.369 | 0.356 | 0.345 | 0.381 | 0.356 | 0.349 | 0.367 | 0.354 | 0.403 | 0.428 | 0.378 | 0.370 | 0.414 | 0.410 | 0.352 | 0.288 |
| | Avg | 0.257 | 0.285 | 0.265 | 0.290 | 0.279 | 0.296 | 0.265 | 0.317 | 0.265 | 0.285 | 0.275 | 0.292 | 0.309 | 0.360 | 0.291 | 0.315 | 0.288 | 0.314 | 0.271 | 0.334 |
| ETTh1 | 96 | 0.403 | 0.420 | 0.414 | 0.431 | 0.414 | 0.422 | 0.386 | 0.400 | 0.460 | 0.447 | 0.409 | 0.415 | 0.376 | 0.419 | 0.411 | 0.426 | 0.513 | 0.491 | 0.494 | 0.479 |
| | 192 | 0.440 | 0.441 | 0.463 | 0.456 | 0.466 | 0.450 | 0.437 | 0.432 | 0.512 | 0.477 | 0.468 | 0.446 | 0.420 | 0.448 | 0.475 | 0.461 | 0.534 | 0.504 | 0.538 | 0.504 |
| | 336 | 0.471 | 0.457 | 0.487 | 0.466 | 0.515 | 0.475 | 0.481 | 0.459 | 0.546 | 0.496 | 0.503 | 0.461 | 0.459 | 0.465 | 0.508 | 0.482 | 0.588 | 0.535 | 0.574 | 0.521 |
| | 720 | 0.503 | 0.487 | 0.517 | 0.494 | 0.503 | 0.487 | 0.519 | 0.516 | 0.544 | 0.517 | 0.510 | 0.482 | 0.506 | 0.507 | 0.504 | 0.494 | 0.643 | 0.616 | 0.562 | 0.535 |
| | Avg | 0.454 | 0.451 | 0.470 | 0.462 | 0.474 | 0.459 | 0.456 | 0.452 | 0.516 | 0.484 | 0.473 | 0.451 | 0.440 | 0.460 | 0.475 | 0.466 | 0.570 | 0.537 | 0.542 | 0.510 |
| ETTh2 | 96 | 0.322 | 0.366 | 0.340 | 0.374 | 0.306 | 0.353 | 0.333 | 0.387 | 0.308 | 0.355 | 0.298 | 0.350 | 0.358 | 0.397 | 0.328 | 0.374 | 0.476 | 0.458 | 0.340 | 0.391 |
| | 192 | 0.400 | 0.409 | 0.399 | 0.410 | 0.386 | 0.399 | 0.477 | 0.476 | 0.393 | 0.405 | 0.376 | 0.399 | 0.429 | 0.439 | 0.403 | 0.418 | 0.512 | 0.493 | 0.430 | 0.439 |
| | 336 | 0.432 | 0.435 | 0.452 | 0.452 | 0.460 | 0.458 | 0.594 | 0.541 | 0.427 | 0.436 | 0.430 | 0.439 | 0.496 | 0.487 | 0.455 | 0.458 | 0.552 | 0.551 | 0.485 | 0.479 |
| | 720 | 0.430 | 0.442 | 0.462 | 0.468 | 0.442 | 0.451 | 0.831 | 0.657 | 0.436 | 0.450 | 0.428 | 0.451 | 0.463 | 0.474 | 0.470 | 0.477 | 0.562 | 0.560 | 0.500 | 0.497 |
| | Avg | 0.396 | 0.413 | 0.413 | 0.426 | 0.398 | 0.415 | 0.559 | 0.515 | 0.391 | 0.411 | 0.383 | 0.410 | 0.437 | 0.449 | 0.414 | 0.432 | 0.526 | 0.516 | 0.439 | 0.452 |
| ETTm1 | 96 | 0.329 | 0.371 | 0.340 | 0.377 | 0.393 | 0.398 | 0.345 | 0.372 | 0.352 | 0.374 | 0.350 | 0.369 | 0.379 | 0.419 | 0.336 | 0.373 | 0.386 | 0.398 | 0.375 | 0.398 |
| | 192 | 0.380 | 0.398 | 0.406 | 0.408 | 0.412 | 0.405 | 0.380 | 0.389 | 0.390 | 0.393 | 0.387 | 0.387 | 0.426 | 0.441 | 0.381 | 0.399 | 0.459 | 0.444 | 0.408 | 0.410 |
| | 336 | 0.418 | 0.425 | 0.424 | 0.425 | 0.442 | 0.425 | 0.413 | 0.413 | 0.421 | 0.414 | 0.418 | 0.407 | 0.445 | 0.459 | 0.411 | 0.418 | 0.495 | 0.464 | 0.435 | 0.428 |
| | 720 | 0.476 | 0.440 | 0.485 | 0.461 | 0.502 | 0.457 | 0.474 | 0.453 | 0.462 | 0.449 | 0.477 | 0.437 | 0.543 | 0.490 | 0.478 | 0.454 | 0.585 | 0.516 | 0.499 | 0.462 |
| | Avg | 0.401 | 0.409 | 0.414 | 0.418 | 0.437 | 0.421 | 0.403 | 0.407 | 0.406 | 0.407 | 0.408 | 0.400 | 0.448 | 0.452 | 0.402 | 0.411 | 0.481 | 0.456 | 0.429 | 0.425 |
| ETTm2 | 96 | 0.189 | 0.266 | 0.185 | 0.264 | 0.193 | 0.281 | 0.193 | 0.292 | 0.183 | 0.270 | 0.185 | 0.271 | 0.203 | 0.287 | 0.230 | 0.307 | 0.192 | 0.274 | 0.189 | 0.280 |
| | 192 | 0.253 | 0.307 | 0.252 | 0.306 | 0.254 | 0.315 | 0.284 | 0.363 | 0.255 | 0.314 | 0.250 | 0.312 | 0.269 | 0.328 | 0.284 | 0.338 | 0.280 | 0.339 | 0.253 | 0.319 |
| | 336 | 0.315 | 0.345 | 0.323 | 0.350 | 0.320 | 0.355 | 0.369 | 0.427 | 0.309 | 0.347 | 0.314 | 0.351 | 0.325 | 0.366 | 0.340 | 0.370 | 0.334 | 0.361 | 0.314 | 0.357 |
| | 720 | 0.421 | 0.408 | 0.415 | 0.403 | 0.426 | 0.416 | 0.554 | 0.522 | 0.412 | 0.404 | 0.410 | 0.408 | 0.421 | 0.415 | 0.436 | 0.420 | 0.417 | 0.413 | 0.414 | 0.413 |
| | Avg | 0.295 | 0.331 | 0.294 | 0.331 | 0.298 | 0.342 | 0.350 | 0.401 | 0.290 | 0.334 | 0.290 | 0.335 | 0.305 | 0.349 | 0.323 | 0.359 | 0.306 | 0.347 | 0.293 | 0.342 |
| ILI | 24 | 1.921 | 0.898 | 2.072 | 0.948 | 2.589 | 1.054 | 2.398 | 1.040 | 2.229 | 0.894 | 5.259 | 1.689 | 3.228 | 1.260 | 3.371 | 1.231 | 2.294 | 0.945 | 2.527 | 1.020 |
| | 36 | 2.151 | 0.933 | 2.494 | 1.019 | 2.996 | 1.194 | 2.646 | 1.088 | 2.330 | 0.925 | 6.136 | 1.831 | 2.679 | 1.080 | 3.725 | 1.322 | 1.825 | 0.848 | 2.615 | 1.007 |
| | 48 | 2.062 | 0.892 | 2.298 | 0.964 | 2.714 | 1.095 | 2.614 | 1.086 | 2.140 | 0.894 | 4.670 | 1.562 | 2.622 | 1.078 | 3.291 | 1.237 | 2.010 | 0.900 | 2.359 | 0.972 |
| | 60 | 1.759 | 0.853 | 2.198 | 0.963 | 2.605 | 1.050 | 2.804 | 1.146 | 2.037 | 0.912 | 4.402 | 1.517 | 2.857 | 1.157 | 2.907 | 1.136 | 2.178 | 0.963 | 2.487 | 1.016 |
| | Avg | 1.973 | 0.894 | 2.266 | 0.974 | 2.726 | 1.098 | 2.616 | 1.090 | 2.184 | 0.906 | 5.117 | 1.650 | 2.847 | 1.144 | 3.324 | 1.232 | 2.077 | 0.914 | 2.497 | 1.004 |
| ECL | 96 | 0.167 | 0.271 | 0.169 | 0.273 | 0.207 | 0.292 | 0.197 | 0.282 | 0.190 | 0.296 | 0.186 | 0.273 | 0.193 | 0.308 | 0.218 | 0.309 | 0.169 | 0.273 | 0.187 | 0.304 |
| | 192 | 0.178 | 0.280 | 0.186 | 0.288 | 0.209 | 0.297 | 0.196 | 0.285 | 0.199 | 0.304 | 0.190 | 0.278 | 0.201 | 0.315 | 0.220 | 0.311 | 0.182 | 0.286 | 0.199 | 0.315 |
| | 336 | 0.198 | 0.302 | 0.206 | 0.305 | 0.224 | 0.312 | 0.209 | 0.301 | 0.217 | 0.319 | 0.204 | 0.291 | 0.214 | 0.329 | 0.234 | 0.323 | 0.200 | 0.304 | 0.212 | 0.329 |
| | 720 | 0.233 | 0.344 | 0.231 | 0.327 | 0.277 | 0.359 | 0.245 | 0.333 | 0.258 | 0.352 | 0.245 | 0.297 | 0.325 | 0.355 | 0.276 | 0.354 | 0.222 | 0.321 | 0.233 | 0.345 |
| | Avg | 0.194 | 0.299 | 0.198 | 0.298 | 0.229 | 0.315 | 0.212 | 0.300 | 0.216 | 0.318 | 0.206 | 0.285 | 0.214 | 0.327 | 0.237 | 0.324 | 0.193 | 0.296 | 0.208 | 0.323 |
| Traffic | 96 | 0.587 | 0.315 | 0.589 | 0.313 | 0.609 | 0.402 | 0.650 | 0.396 | 0.526 | 0.347 | 0.563 | 0.378 | 0.587 | 0.366 | 0.589 | 0.390 | 0.612 | 0.338 | 0.607 | 0.392 |
| | 192 | 0.612 | 0.326 | 0.627 | 0.337 | 0.586 | 0.382 | 0.598 | 0.370 | 0.522 | 0.332 | 0.549 | 0.367 | 0.604 | 0.373 | 0.567 | 0.380 | 0.613 | 0.340 | 0.621 | 0.399 |
| | 336 | 0.634 | 0.338 | 0.635 | 0.341 | 0.593 | 0.390 | 0.605 | 0.373 | 0.517 | 0.334 | 0.566 | 0.376 | 0.621 | 0.383 | 0.583 | 0.389 | 0.618 | 0.328 | 0.622 | 0.396 |
| | 720 | 0.640 | 0.351 | 0.658 | 0.349 | 0.636 | 0.405 | 0.645 | 0.394 | 0.552 | 0.352 | 0.567 | 0.372 | 0.626 | 0.382 | 0.585 | 0.391 | 0.653 | 0.355 | 0.632 | 0.396 |
| | Avg | 0.618 | 0.333 | 0.627 | 0.335 | 0.606 | 0.395 | 0.625 | 0.383 | 0.529 | 0.341 | 0.561 | 0.373 | 0.610 | 0.376 | 0.581 | 0.388 | 0.624 | 0.340 | 0.621 | 0.396 |
| Average | | 0.574 | 0.427 | 0.618 | 0.442 | 0.681 | 0.468 | 0.686 | 0.483 | 0.600 | 0.436 | 0.964 | 0.525 | 0.701 | 0.489 | 0.756 | 0.491 | 0.633 | 0.465 | 0.662 | 0.473 |

**Sample Reweighting Illustration.** Figures 11, 10, 12, and 13 display the learned weighting network applied to various datasets: MSL for anomaly detection, Weather for forecasting, ETTh1 for imputation, and PEMS-SF for classification. These visualizations corroborate our hypothesis: the sample weight $\omega_O$ increases with the prediction loss $l_O$, while the weight $\omega_I$ decreases as $l_O$ increases. This observed pattern supports the efficacy of our reweighting strategy.

**Static Weighting Scheme.** We also explore a static weighting scheme as a contrast to the dynamic weighting used in our sample reweighting module. This scheme balances the prediction loss and mutual information loss, with a ratio of 0.0 representing pure prediction loss and 1.0 representing pure mutual information loss. As shown in Table 18, the static approach underperforms relative to our dynamic sample weighting module, demonstrating the superior effectiveness of our method.

### A.12.1 Comprehensive Results.

The detailed performance of various traditional TS models and LLMs is presented in Table 16 and Table 17.

Table 13: Full results for the imputation task. Randomly masked {12.5%, 25%, 37.5%, 50%} of points in 96-length series, averaging results over 4 mask ratios.

| Methods Mask | Ratio | LLM-TS MSE | MAE | TimesNet MSE | MAE | GPT4TS MSE | MAE | PatchTST MSE | MAE | LightTS MSE | MAE | DLinear MSE | MAE | FEDformer MSE | MAE | Stationary MSE | MAE | Autoformer MSE | MAE | Reformer MSE | MAE |
|---|---|---|---|---|---|---|---|---|---|---|---|---|---|---|---|---|---|---|---|---|---|
| ETTm1 | 12.5% | 0.018 | 0.088 | 0.023 | 0.101 | 0.018 | 0.089 | 0.041 | 0.130 | 0.093 | 0.206 | 0.080 | 0.193 | 0.052 | 0.166 | 0.032 | 0.119 | 0.046 | 0.144 | 0.042 | 0.146 |
|  | 25% | 0.022 | 0.097 | 0.023 | 0.101 | 0.023 | 0.099 | 0.044 | 0.135 | 0.093 | 0.206 | 0.080 | 0.193 | 0.052 | 0.166 | 0.032 | 0.119 | 0.046 | 0.144 | 0.042 | 0.146 |
|  | 37.5% | 0.027 | 0.108 | 0.029 | 0.112 | 0.030 | 0.112 | 0.049 | 0.143 | 0.113 | 0.231 | 0.103 | 0.219 | 0.069 | 0.191 | 0.039 | 0.131 | 0.057 | 0.161 | 0.063 | 0.182 |
|  | 50% | 0.033 | 0.120 | 0.035 | 0.123 | 0.042 | 0.131 | 0.055 | 0.151 | 0.134 | 0.255 | 0.132 | 0.248 | 0.089 | 0.218 | 0.047 | 0.145 | 0.067 | 0.174 | 0.082 | 0.208 |
|  | Avg | 0.025 | 0.103 | 0.028 | 0.109 | 0.028 | 0.108 | 0.047 | 0.140 | 0.104 | 0.218 | 0.093 | 0.206 | 0.062 | 0.177 | 0.036 | 0.126 | 0.051 | 0.150 | 0.055 | 0.166 |
| ETTm2 | 12.5% | 0.018 | 0.079 | 0.019 | 0.081 | 0.019 | 0.078 | 0.108 | 0.239 | 0.034 | 0.127 | 0.062 | 0.166 | 0.056 | 0.159 | 0.021 | 0.088 | 0.023 | 0.092 | 0.108 | 0.228 |
|  | 25% | 0.020 | 0.085 | 0.021 | 0.087 | 0.021 | 0.084 | 0.028 | 0.099 | 0.042 | 0.143 | 0.085 | 0.196 | 0.080 | 0.195 | 0.024 | 0.096 | 0.026 | 0.101 | 0.136 | 0.262 |
|  | 37.5% | 0.022 | 0.089 | 0.023 | 0.092 | 0.024 | 0.090 | 0.030 | 0.104 | 0.051 | 0.159 | 0.106 | 0.222 | 0.110 | 0.231 | 0.027 | 0.103 | 0.030 | 0.108 | 0.175 | 0.300 |
|  | 50% | 0.025 | 0.096 | 0.025 | 0.097 | 0.027 | 0.098 | 0.034 | 0.110 | 0.059 | 0.174 | 0.131 | 0.247 | 0.156 | 0.276 | 0.030 | 0.108 | 0.035 | 0.119 | 0.211 | 0.329 |
|  | Avg | 0.021 | 0.087 | 0.022 | 0.089 | 0.023 | 0.088 | 0.029 | 0.102 | 0.046 | 0.151 | 0.096 | 0.208 | 0.101 | 0.215 | 0.026 | 0.099 | 0.029 | 0.105 | 0.157 | 0.280 |
| ETTh1 | 12.5% | 0.058 | 0.165 | 0.064 | 0.170 | 0.043 | 0.141 | 0.093 | 0.201 | 0.240 | 0.345 | 0.151 | 0.267 | 0.070 | 0.190 | 0.060 | 0.165 | 0.074 | 0.182 | 0.074 | 0.194 |
|  | 25% | 0.077 | 0.189 | 0.082 | 0.192 | 0.056 | 0.159 | 0.107 | 0.217 | 0.265 | 0.364 | 0.180 | 0.292 | 0.106 | 0.236 | 0.080 | 0.189 | 0.090 | 0.203 | 0.102 | 0.227 |
|  | 37.5% | 0.096 | 0.209 | 0.098 | 0.209 | 0.074 | 0.182 | 0.120 | 0.230 | 0.296 | 0.382 | 0.215 | 0.318 | 0.124 | 0.258 | 0.102 | 0.212 | 0.109 | 0.222 | 0.135 | 0.261 |
|  | 50% | 0.118 | 0.228 | 0.116 | 0.226 | 0.104 | 0.214 | 0.141 | 0.248 | 0.334 | 0.404 | 0.257 | 0.347 | 0.165 | 0.299 | 0.133 | 0.240 | 0.137 | 0.248 | 0.179 | 0.298 |
|  | Avg | 0.087 | 0.198 | 0.090 | 0.199 | 0.069 | 0.174 | 0.115 | 0.224 | 0.284 | 0.373 | 0.201 | 0.306 | 0.117 | 0.246 | 0.094 | 0.201 | 0.103 | 0.214 | 0.122 | 0.245 |
| ETTh2 | 12.5% | 0.039 | 0.131 | 0.040 | 0.132 | 0.041 | 0.129 | 0.057 | 0.152 | 0.101 | 0.231 | 0.100 | 0.216 | 0.095 | 0.212 | 0.042 | 0.133 | 0.044 | 0.138 | 0.163 | 0.289 |
|  | 25% | 0.046 | 0.143 | 0.048 | 0.146 | 0.046 | 0.137 | 0.061 | 0.158 | 0.115 | 0.246 | 0.127 | 0.247 | 0.137 | 0.258 | 0.049 | 0.147 | 0.050 | 0.149 | 0.206 | 0.331 |
|  | 37.5% | 0.053 | 0.154 | 0.055 | 0.156 | 0.053 | 0.148 | 0.067 | 0.166 | 0.126 | 0.257 | 0.158 | 0.276 | 0.187 | 0.304 | 0.056 | 0.158 | 0.060 | 0.163 | 0.252 | 0.370 |
|  | 50% | 0.061 | 0.165 | 0.061 | 0.165 | 0.060 | 0.160 | 0.073 | 0.174 | 0.136 | 0.268 | 0.183 | 0.299 | 0.232 | 0.341 | 0.065 | 0.170 | 0.068 | 0.173 | 0.316 | 0.419 |
|  | Avg | 0.050 | 0.148 | 0.051 | 0.150 | 0.050 | 0.144 | 0.065 | 0.163 | 0.119 | 0.250 | 0.142 | 0.259 | 0.163 | 0.279 | 0.053 | 0.152 | 0.055 | 0.156 | 0.234 | 0.352 |
| ECL | 12.5% | 0.087 | 0.203 | 0.090 | 0.204 | 0.080 | 0.194 | 0.055 | 0.160 | 0.102 | 0.229 | 0.092 | 0.214 | 0.107 | 0.237 | 0.093 | 0.210 | 0.089 | 0.210 | 0.190 | 0.308 |
|  | 25% | 0.091 | 0.207 | 0.092 | 0.209 | 0.087 | 0.203 | 0.065 | 0.175 | 0.121 | 0.252 | 0.118 | 0.247 | 0.120 | 0.251 | 0.097 | 0.214 | 0.096 | 0.220 | 0.197 | 0.312 |
|  | 37.5% | 0.095 | 0.213 | 0.096 | 0.213 | 0.094 | 0.211 | 0.076 | 0.344 | 0.141 | 0.273 | 0.144 | 0.276 | 0.136 | 0.266 | 0.102 | 0.220 | 0.104 | 0.229 | 0.203 | 0.315 |
|  | 50% | 0.101 | 0.220 | 0.102 | 0.221 | 0.101 | 0.220 | 0.091 | 0.208 | 0.160 | 0.293 | 0.175 | 0.305 | 0.158 | 0.284 | 0.108 | 0.228 | 0.113 | 0.239 | 0.210 | 0.319 |
|  | Avg | 0.094 | 0.211 | 0.095 | 0.212 | 0.091 | 0.207 | 0.072 | 0.183 | 0.131 | 0.262 | 0.132 | 0.260 | 0.130 | 0.259 | 0.100 | 0.218 | 0.101 | 0.225 | 0.200 | 0.313 |
| Weather | 12.5% | 0.026 | 0.048 | 0.025 | 0.047 | 0.027 | 0.049 | 0.029 | 0.049 | 0.047 | 0.101 | 0.039 | 0.084 | 0.041 | 0.107 | 0.027 | 0.051 | 0.026 | 0.047 | 0.031 | 0.076 |
|  | 25% | 0.029 | 0.055 | 0.031 | 0.062 | 0.030 | 0.054 | 0.031 | 0.053 | 0.052 | 0.111 | 0.048 | 0.103 | 0.064 | 0.163 | 0.029 | 0.056 | 0.030 | 0.054 | 0.035 | 0.082 |
|  | 37.5% | 0.032 | 0.059 | 0.034 | 0.064 | 0.034 | 0.062 | 0.035 | 0.058 | 0.058 | 0.121 | 0.057 | 0.117 | 0.107 | 0.229 | 0.033 | 0.062 | 0.032 | 0.060 | 0.040 | 0.091 |
|  | 50% | 0.033 | 0.061 | 0.035 | 0.062 | 0.037 | 0.066 | 0.038 | 0.063 | 0.065 | 0.133 | 0.066 | 0.134 | 0.183 | 0.312 | 0.037 | 0.068 | 0.037 | 0.067 | 0.046 | 0.099 |
|  | Avg | 0.030 | 0.056 | 0.031 | 0.059 | 0.032 | 0.058 | 0.060 | 0.144 | 0.055 | 0.117 | 0.052 | 0.110 | 0.099 | 0.203 | 0.032 | 0.059 | 0.031 | 0.057 | 0.038 | 0.087 |

Table 14: Complete classification task results. *. in the Transformers indicates the name of *former.

| Methods | Classical XGB | Roc | RNN LSTNet | LSSL | TCN | Trans. | Re. | In. | Pyra. | Auto. | Station. | FED. | ETS. | Flow. | MLP DL | LTS. | TimesNet | LLM GPT4TS | TEST | LLM-TS |
|---|---|---|---|---|---|---|---|---|---|---|---|---|---|---|---|---|---|---|---|---|
| Ethanol | 43.7 | 45.2 | 39.9 | 31.1 | 28.9 | 32.7 | 31.9 | 31.6 | 30.8 | 31.6 | 32.7 | 31.2 | 28.1 | 33.8 | 32.6 | 29.7 | 30.4 | 26.2 | 25.1 | 31.9 |
| FaceD | 63.3 | 64.7 | 65.7 | 66.7 | 52.8 | 67.3 | 68.6 | 67.0 | 65.7 | 68.4 | 68.0 | 66.0 | 66.3 | 67.6 | 68.0 | 67.5 | 68.6 | 67.8 | 50.1 | 68.9 |
| HandW | 15.8 | 58.8 | 25.8 | 24.6 | 53.3 | 32.0 | 27.4 | 32.8 | 29.4 | 36.7 | 31.6 | 28.0 | 32.5 | 33.8 | 27.0 | 26.1 | 32.1 | 28.9 | 20.1 | 32.7 |
| HeartB | 73.2 | 75.6 | 77.1 | 72.7 | 75.6 | 76.1 | 77.1 | 80.5 | 75.6 | 74.6 | 73.7 | 73.7 | 71.2 | 77.6 | 75.1 | 75.1 | 77.6 | 72.2 | 73.7 | 77.1 |
| JapanV | 86.5 | 96.2 | 98.1 | 98.4 | 98.9 | 98.7 | 97.8 | 98.9 | 98.4 | 96.2 | 99.2 | 98.4 | 95.9 | 98.9 | 96.2 | 96.2 | 97.2 | 98.4 | 78.4 | 98.1 |
| PEMS | 98.3 | 75.1 | 86.7 | 86.1 | 68.8 | 82.1 | 82.7 | 81.5 | 83.2 | 82.7 | 87.3 | 80.9 | 86.0 | 83.8 | 75.1 | 88.4 | 89.6 | 79.2 | 59.5 | 90.8 |
| SCP1 | 84.6 | 90.8 | 84.0 | 90.8 | 84.6 | 92.2 | 90.4 | 90.1 | 88.1 | 84.0 | 89.4 | 88.7 | 89.6 | 92.5 | 87.3 | 89.8 | 90.4 | 90.1 | 84.0 | 91.8 |
| SCP2 | 48.9 | 53.3 | 52.8 | 52.2 | 55.6 | 53.9 | 56.7 | 53.3 | 53.3 | 50.6 | 57.2 | 54.4 | 55.0 | 56.1 | 50.5 | 51.1 | 57.1 | 50.0 | 54.4 | 57.8 |
| SpokenA | 69.6 | 71.2 | 100.0 | 100.0 | 95.6 | 98.4 | 97.0 | 100.0 | 99.6 | 100.0 | 100.0 | 100.0 | 100.0 | 98.8 | 81.4 | 100.0 | 98.6 | 97.9 | 82.1 | 98.6 |
| UWave | 75.9 | 94.4 | 87.8 | 85.9 | 88.4 | 85.6 | 85.6 | 85.6 | 83.4 | 85.9 | 87.5 | 85.3 | 85.0 | 86.6 | 82.1 | 80.3 | 85.5 | 85.6 | 84.4 | 86.6 |
| Avg | 66.0 | 72.5 | 71.8 | 70.9 | 70.3 | 71.9 | 71.5 | 72.1 | 70.8 | 71.1 | 72.7 | 70.7 | 71.0 | 73.0 | 67.5 | 70.4 | 72.7 | 69.5 | 61.2 | **73.4** |

Table 15: Full results for the anomaly detection.

| Methods Metrics | SMD P | R | F1 | MSL P | R | F1 | SMAP P | R | F1 | SWaT P | R | F1 | PSM P | R | F1 | Avg F1 % |
|---|---|---|---|---|---|---|---|---|---|---|---|---|---|---|---|---|
| LLM-TS | 88.09 | 81.54 | 84.69 | 89.04 | 74.49 | 81.11 | 89.95 | 56.51 | 69.41 | 91.16 | 95.40 | 93.23 | 98.44 | 96.45 | 97.43 | 85.17 |
| TimesNet | 87.93 | 81.45 | 84.57 | 88.62 | 73.48 | 80.34 | 89.59 | 56.35 | 69.18 | 91.00 | 95.33 | 93.12 | 98.40 | 96.18 | 97.27 | 84.90 |
| GPT4TS | 87.70 | 81.19 | 84.32 | 82.15 | 81.32 | 81.73 | 90.04 | 55.75 | 68.86 | 92.12 | 93.06 | 92.59 | 98.37 | 96.34 | 97.34 | 84.97 |
| PatchTST | 87.26 | 82.14 | 84.62 | 88.34 | 70.96 | 78.70 | 90.64 | 55.46 | 68.82 | 91.10 | 80.94 | 85.72 | 98.84 | 93.47 | 96.08 | 82.79 |
| ETSformer | 87.44 | 79.23 | 83.13 | 85.13 | 84.93 | 85.03 | 92.25 | 55.75 | 69.50 | 90.02 | 80.36 | 84.91 | 99.31 | 85.28 | 91.76 | 82.87 |
| FEDformer | 87.95 | 82.39 | 85.08 | 77.14 | 80.07 | 78.57 | 90.47 | 58.10 | 70.76 | 90.17 | 96.42 | 93.19 | 97.31 | 97.16 | 97.23 | 84.97 |
| LightTS | 87.10 | 78.42 | 82.53 | 82.40 | 75.78 | 78.95 | 92.58 | 55.27 | 69.21 | 91.98 | 94.72 | 93.33 | 98.37 | 95.97 | 97.15 | 84.23 |
| DLinear | 83.62 | 71.52 | 77.10 | 84.34 | 85.42 | 84.88 | 92.32 | 55.41 | 69.26 | 80.91 | 95.30 | 87.52 | 98.28 | 89.26 | 93.55 | 82.46 |
| Stationary | 88.33 | 81.21 | 84.62 | 68.55 | 89.14 | 77.50 | 89.37 | 59.02 | 71.09 | 68.03 | 96.75 | 79.88 | 97.82 | 96.76 | 97.29 | 82.08 |
| Autoformer | 88.06 | 82.35 | 85.11 | 77.27 | 80.92 | 79.05 | 90.40 | 58.62 | 71.12 | 89.85 | 95.81 | 92.74 | 99.08 | 88.15 | 93.29 | 84.26 |
| Pyraformer | 85.61 | 80.61 | 83.04 | 83.81 | 85.93 | 84.86 | 92.54 | 57.71 | 71.09 | 87.92 | 96.00 | 91.78 | 71.67 | 96.02 | 82.08 | 82.57 |
| Anomaly Transformer | 88.91 | 82.23 | 85.49 | 79.61 | 87.37 | 83.31 | 91.85 | 58.11 | 71.18 | 72.51 | 97.32 | 83.10 | 68.35 | 94.72 | 79.40 | 80.50 |
| Informer | 86.60 | 77.23 | 81.65 | 81.77 | 86.48 | 84.06 | 90.11 | 57.13 | 69.92 | 70.29 | 96.75 | 81.43 | 64.27 | 96.33 | 77.10 | 78.83 |
| Reformer | 82.58 | 69.24 | 75.32 | 85.51 | 83.31 | 84.40 | 90.91 | 57.44 | 70.40 | 72.50 | 96.53 | 82.80 | 59.93 | 95.38 | 73.61 | 77.31 |
| Transformer | 83.58 | 76.13 | 79.56 | 71.57 | 87.37 | 78.68 | 89.37 | 57.12 | 69.70 | 68.84 | 96.53 | 80.37 | 62.75 | 96.56 | 76.07 | 76.88 |

Table 16: Different traditional models. We use prediction length $O \in \{96, 192, 336, 720\}$ for ILI and $O \in \{24, 36, 48, 60\}$ for others.

| Methods | | PatchTST | | PatchTST INT | | ETSformer | | ETS INT | | Stationary | | Stat INT | | FreTS | | FreTS INT | |
|---|---|---|---|---|---|---|---|---|---|---|---|---|---|---|---|---|---|---|
| Metric | | MSE | MAE | MSE | MAE | MSE | MAE | MSE | MAE | MSE | MAE | MSE | MAE | MSE | MAE | MSE | MAE |
| *Weather* | 96 | 0.174 | 0.216 | 0.172 | 0.214 | 0.196 | 0.282 | 0.200 | 0.285 | 0.178 | 0.226 | 0.201 | 0.246 | 0.187 | 0.243 | 0.179 | 0.235 |
| | 192 | 0.222 | 0.258 | 0.219 | 0.255 | 0.282 | 0.364 | 0.278 | 0.361 | 0.235 | 0.278 | 0.238 | 0.280 | 0.227 | 0.274 | 0.221 | 0.278 |
| | 336 | 0.280 | 0.298 | 0.279 | 0.298 | 0.344 | 0.409 | 0.322 | 0.382 | 0.327 | 0.339 | 0.312 | 0.329 | 0.281 | 0.325 | 0.276 | 0.320 |
| | 720 | 0.356 | 0.349 | 0.356 | 0.348 | 0.430 | 0.472 | 0.427 | 0.470 | 0.387 | 0.383 | 0.386 | 0.383 | 0.352 | 0.382 | 0.344 | 0.376 |
| | Avg | 0.258 | 0.280 | 0.257 | 0.279 | 0.313 | 0.382 | 0.307 | 0.375 | 0.282 | 0.307 | 0.284 | 0.309 | 0.262 | 0.306 | 0.255 | 0.302 |
| *ETTh1* | 96 | 0.381 | 0.398 | 0.382 | 0.401 | 0.554 | 0.536 | 0.550 | 0.532 | 0.534 | 0.499 | 0.523 | 0.486 | 0.398 | 0.412 | 0.395 | 0.409 |
| | 192 | 0.421 | 0.426 | 0.422 | 0.428 | 0.686 | 0.619 | 0.690 | 0.621 | 0.639 | 0.560 | 0.609 | 0.560 | 0.454 | 0.449 | 0.455 | 0.451 |
| | 336 | 0.464 | 0.449 | 0.460 | 0.441 | 0.869 | 0.730 | 0.868 | 0.728 | 0.790 | 0.648 | 0.780 | 0.634 | 0.512 | 0.483 | 0.502 | 0.474 |
| | 720 | 0.527 | 0.500 | 0.510 | 0.496 | 1.085 | 0.849 | 1.054 | 0.830 | 0.706 | 0.620 | 0.701 | 0.606 | 0.572 | 0.547 | 0.560 | 0.530 |
| | Avg | 0.448 | 0.443 | 0.444 | 0.442 | 0.799 | 0.684 | 0.791 | 0.678 | 0.667 | 0.582 | 0.653 | 0.572 | 0.484 | 0.473 | 0.478 | 0.466 |
| *ETTm1* | 96 | 0.332 | 0.368 | 0.332 | 0.372 | 0.526 | 0.495 | 0.424 | 0.434 | 0.417 | 0.417 | 0.412 | 0.410 | 0.340 | 0.375 | 0.339 | 0.374 |
| | 192 | 0.368 | 0.388 | 0.367 | 0.388 | 0.565 | 0.538 | 0.458 | 0.461 | 0.446 | 0.437 | 0.445 | 0.435 | 0.395 | 0.408 | 0.384 | 0.399 |
| | 336 | 0.397 | 0.405 | 0.396 | 0.405 | 0.658 | 0.603 | 0.537 | 0.519 | 0.582 | 0.507 | 0.570 | 0.491 | 0.431 | 0.433 | 0.420 | 0.423 |
| | 720 | 0.457 | 0.445 | 0.460 | 0.446 | 0.801 | 0.696 | 0.802 | 0.696 | 0.661 | 0.546 | 0.660 | 0.546 | 0.494 | 0.470 | 0.484 | 0.462 |
| | Avg | 0.389 | 0.402 | 0.389 | 0.403 | 0.638 | 0.583 | 0.555 | 0.528 | 0.527 | 0.477 | 0.522 | 0.471 | 0.415 | 0.422 | 0.407 | 0.415 |
| *ILI* | 24 | 2.229 | 0.894 | 2.172 | 0.856 | 4.043 | 1.410 | 3.607 | 1.305 | 2.722 | 1.024 | 1.905 | 0.872 | 3.226 | 1.231 | 3.202 | 1.213 |
| | 36 | 2.330 | 0.925 | 2.347 | 0.978 | 3.809 | 1.358 | 3.705 | 1.315 | 3.026 | 1.071 | 2.790 | 1.068 | 3.363 | 1.259 | 3.000 | 1.173 |
| | 48 | 2.140 | 0.894 | 1.984 | 0.869 | 3.851 | 1.351 | 3.714 | 1.309 | 2.622 | 1.032 | 2.132 | 0.900 | 3.456 | 1.285 | 3.132 | 1.213 |
| | 60 | 2.037 | 0.912 | 1.770 | 0.831 | 3.983 | 1.349 | 3.935 | 1.350 | 2.520 | 1.035 | 1.991 | 0.901 | 3.749 | 1.340 | 3.298 | 1.243 |
| | Avg | 2.184 | 0.906 | 2.068 | 0.884 | 3.922 | 1.367 | 3.740 | 1.320 | 2.722 | 1.041 | 2.205 | 0.935 | 3.449 | 1.279 | 3.158 | 1.211 |

Table 17: Different LLM embeddings. We use prediction length $O \in \{96, 192, 336, 720\}$ for ILI and $O \in \{24, 36, 48, 60\}$ for others.

| Methods | | LLM-TS (LLaMA) | | LLaMA w/o text | | GPT2 | | BERT | | No LLM | |
|---|---|---|---|---|---|---|---|---|---|---|---|
| Metric | | MSE | MAE | MSE | MAE | MSE | MAE | MSE | MAE | MSE | MAE |
| *Weather* | 96 | 0.166 | 0.217 | 0.170 | 0.218 | 0.168 | 0.218 | 0.167 | 0.217 | 0.168 | 0.218 |
| | 192 | 0.229 | 0.269 | 0.227 | 0.266 | 0.226 | 0.267 | 0.229 | 0.270 | 0.227 | 0.268 |
| | 336 | 0.278 | 0.302 | 0.295 | 0.314 | 0.292 | 0.310 | 0.283 | 0.305 | 0.298 | 0.318 |
| | 720 | 0.354 | 0.351 | 0.360 | 0.354 | 0.359 | 0.354 | 0.360 | 0.354 | 0.361 | 0.356 |
| | Avg | 0.257 | 0.285 | 0.263 | 0.288 | 0.261 | 0.287 | 0.260 | 0.287 | 0.264 | 0.290 |
| *ETTh1* | 96 | 0.403 | 0.420 | 0.409 | 0.427 | 0.408 | 0.426 | 0.402 | 0.421 | 0.402 | 0.422 |
| | 192 | 0.440 | 0.441 | 0.445 | 0.445 | 0.442 | 0.444 | 0.452 | 0.450 | 0.459 | 0.455 |
| | 336 | 0.471 | 0.457 | 0.490 | 0.472 | 0.487 | 0.467 | 0.494 | 0.472 | 0.471 | 0.457 |
| | 720 | 0.503 | 0.487 | 0.518 | 0.496 | 0.517 | 0.494 | 0.520 | 0.497 | 0.535 | 0.507 |
| | Avg | 0.454 | 0.451 | 0.465 | 0.460 | 0.464 | 0.458 | 0.467 | 0.460 | 0.467 | 0.460 |
| *ETTm1* | 96 | 0.329 | 0.371 | 0.350 | 0.387 | 0.338 | 0.370 | 0.340 | 0.375 | 0.341 | 0.377 |
| | 192 | 0.380 | 0.398 | 0.383 | 0.398 | 0.392 | 0.404 | 0.401 | 0.408 | 0.404 | 0.413 |
| | 336 | 0.418 | 0.425 | 0.423 | 0.426 | 0.416 | 0.423 | 0.414 | 0.421 | 0.432 | 0.428 |
| | 720 | 0.476 | 0.440 | 0.467 | 0.449 | 0.477 | 0.454 | 0.470 | 0.445 | 0.468 | 0.449 |
| | Avg | 0.401 | 0.409 | 0.406 | 0.415 | 0.406 | 0.413 | 0.406 | 0.412 | 0.411 | 0.417 |
| *ILI* | 24 | 1.921 | 0.898 | 1.998 | 0.929 | 1.997 | 0.929 | 1.917 | 0.915 | 2.170 | 0.947 |
| | 36 | 2.151 | 0.933 | 2.422 | 0.957 | 2.333 | 0.958 | 2.431 | 1.004 | 2.093 | 0.889 |
| | 48 | 2.062 | 0.892 | 2.198 | 0.964 | 2.269 | 0.937 | 2.333 | 0.961 | 2.418 | 0.959 |
| | 60 | 1.759 | 0.853 | 2.072 | 0.948 | 2.077 | 0.921 | 2.089 | 0.926 | 2.203 | 0.971 |
| | Avg | 1.973 | 0.894 | 2.173 | 0.950 | 2.169 | 0.936 | 2.193 | 0.952 | 2.221 | 0.942 |

Table 18: Static Weighting Scheme with Different ratios.

| Ratio | | 0.0 | | 0.2 | | 0.4 | | 0.6 | | 0.8 | | 1.0 | | Ours | |
|---|---|---|---|---|---|---|---|---|---|---|---|---|---|---|---|
| Metric | | MSE | MAE | MSE | MAE | MSE | MAE | MSE | MAE | MSE | MAE | MSE | MAE | MSE | MAE |
| *ETTh1* | | 0.478 | 0.468 | 0.471 | 0.459 | 0.465 | 0.462 | 0.470 | 0.463 | 0.473 | 0.450 | 0.471 | 0.463 | 0.454 | 0.451 |
| *ETTm1* | | 0.415 | 0.417 | 0.408 | 0.414 | 0.405 | 0.412 | 0.406 | 0.412 | 0.417 | 0.419 | 0.416 | 0.419 | 0.401 | 0.409 |

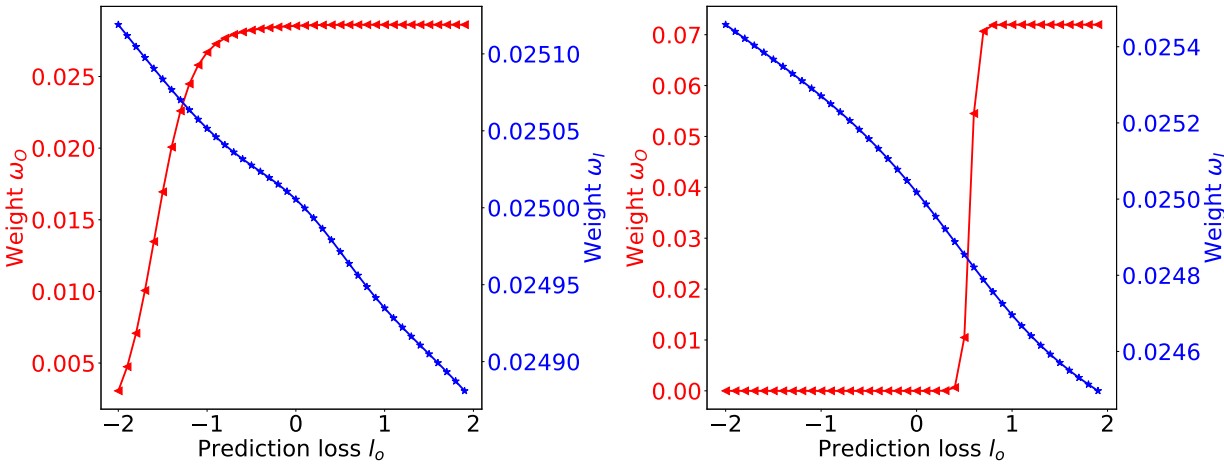

Figure 10: Forecasting.

Figure 11: Anomaly detection

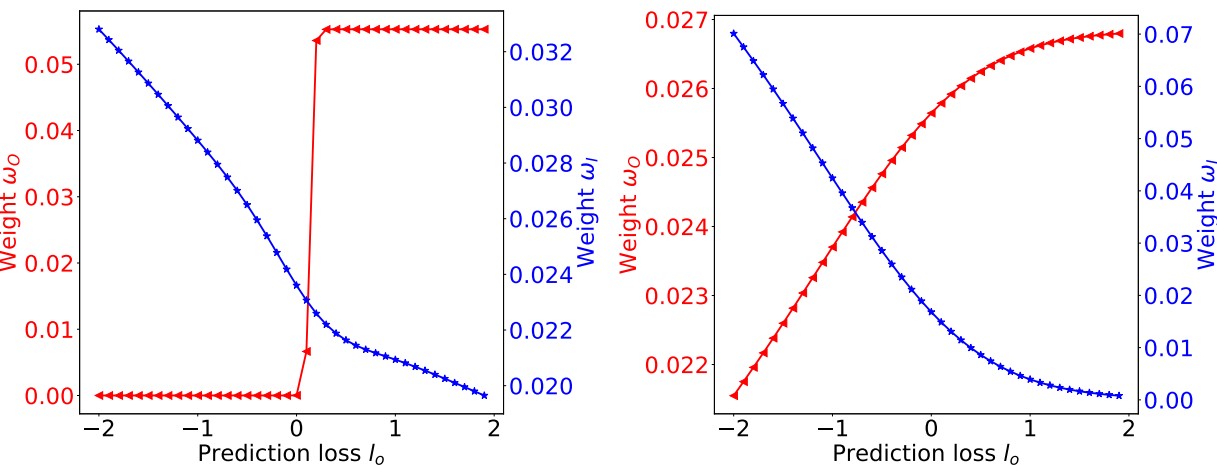

Figure 12: imputation.

Figure 13: classification

