# OpenReview forum: "LLM-TS Integrator: Integrating LLM for Enhanced Time Series Modeling"
_TMLR — Accepted by TMLR_

### Review · Reviewer_zRNN · 2025-02-06

**Summary Of Contributions:**

The paper proposes a generic technique for improving the accuracy of time series models by integrating side information from LLMs. The proposed method consists of two components. First, a mutual information (MI) based loss is used to maximize the MI between the embedding of the TS model and the LLM that processes a textual description of the task. Next, a weighting module is used to dynamically balance the contribution of the MI loss and the prediction loss of the TS model. The proposed framework can be combined with any TS model that produces intermediate embeddings of the time series. Experimental evaluation on forecasting, anomaly detection, imputation and classification tasks demonstrates the consistent improvements produced by the LLM-TS framework compared to the original TimesNet model.

**Audience:**

Yes

**Claims And Evidence:**

Yes

**Requested Changes:**

Please clarify the following aspects in the revised version of the manuscript.

**Critical:**

1. Please include the results for the DHR-ARIMA baseline from [Hewamalage et al., 2023](https://link.springer.com/article/10.1007/s10618-022-00894-5) to the tables showcasing the forecasting performance (Section 4.2)
2. Why do the ablation experiments in Section 4.6 only consider 4 datasets (Weather, ETTh1, ETTm1, ILI) and omit the 4 datasets (ETTh2, ETTm2, ECL, Traffic) used in previous experiments?
3. How is the ablation "w/o template" performed?

**Nice to have:**

4. Section 3.1.1: How is the discriminator $T_\beta$ defined?
5. Section 3.2: Please clarify the meaning of the prediction loss $l_O$ used in different tasks.
6. Why are none of the related models described in Section 5.2 (e.g., Time-LLM, LLMTIME) are not considered in the evaluation? Just a description in the text would be sufficient.
7. Given that the template used in this work only contains simple summary statistics of the time series and the original TS values, it would be interesting to investigate whether the LLM component is necessary at all. As an ablation, one could replace the LLM with an MLP that takes an input the appropriately normalized concatenated statistics rrom the template and outputs the embedding.

**Strengths And Weaknesses:**

**Strengths:**
- The proposed approach provides consistent improvements to the prediction accuracy compared to the original forecasting model (TimesNet). The method can, in principle, be combined with arbitrary TS models that produce an intermediate embedding, which is a big plus.
- The paper is well written and easy to follow. The proposed method is presented very clearly and the relevant concepts are introduced, making the paper accessible to readers with different backgrounds.
- The experimental evaluation includes a large number of ablation studies that investigate the effect of different model components on the final performance.

**Weaknesses:**
- The empirical evaluation covers a large number of tasks but primarily focuses on a handful of benchmark datasets that have been shown to have serious limitations, as well as omits relevant baselines such as statistical models (see, e.g., Table 4 / Section 3.1.2 in [Hewamalage et al., 2023](https://link.springer.com/article/10.1007/s10618-022-00894-5)). I understand that at this point the authors are unable to re-run all the experiments using different benchmarks, but I would urge them to consider alternatives such as the [Monash repository](https://arxiv.org/abs/2105.06643) or [GIFT-Eval](https://arxiv.org/abs/2410.10393) in the future.
- The visualizations of forecasts produced by baseline models shown in Figures 7 and 8 look very disturbing. The forecasts are significantly less accurate than, e.g., a seasonal naive forecast, which could indicate some serious problems with the implementation of the baselines.
- There are several minor points related to the presentation that need to clarified / improved. Please see "Requested changes"

---

> ### Author Response · Authors · 2025-03-07
> **Rebuttal 1/2**
>
> ## General Reply
>
> We sincerely appreciate your insightful feedback, which has greatly improved our paper. In response to your suggestions, we have revised the manuscript accordingly, with all changes highlighted in red for your convenience.
>
> ## Weaknesses
>
> > The empirical evaluation covers a large number of tasks but primarily focuses on a handful of benchmark datasets that have been shown to have serious limitations, as well as omits relevant baselines such as statistical models (see, e.g., Table 4 / Section 3.1.2 in Hewamalage et al., 2023). I understand that at this point the authors are unable to re-run all the experiments using different benchmarks, but I would urge them to consider alternatives such as the Monash repository or GIFT-Eval in the future.
>
> We agree that expanding our evaluation to include more diverse and robust datasets is important. In our revised manuscript, we have added the following sentence in Section 6 Conclusion and Discussion:
>
> > further research is necessary to confirm these results across more diverse and robust datasets including Monash and GIFT-Eval
>
> This addition underscores our commitment to exploring these datasets in future work.
>
>
> > The visualizations of forecasts produced by baseline models shown in Figures 7 and 8 look very disturbing. The forecasts are significantly less accurate than, e.g., a seasonal naive forecast, which could indicate some serious problems with the implementation of the baselines.
>
> We appreciate the reviewer's observation regarding the visualizations in Figures 7 and 8. We would like to clarify and address this concern explicitly:
>
> 1. **Baseline Implementation**: We confirm that our baseline implementations strictly follow the official, standard versions as established in the existing literature.
>
> 2. **Validation of Baseline Implementation**: In Figures 5 (ETTh1) and 6 (ETTm1) of our paper, both the baselines and our proposed method produce accurate and reasonable forecasts. These figures clearly demonstrate that the baseline methods are correctly implemented and function as expected under typical distribution conditions.
>
> 3. **Explanation for Figures 7 and 8**: Figures 7 and 8 specifically illustrate scenarios deliberately constructed to highlight a known failure mode of existing baseline methods: generalization to out-of-distribution settings, where test datasets contain higher-frequency components not observed during training. The reduced accuracy seen in baseline forecasts, demonstrates significantly improved generalization and robustness of our method in such challenging scenarios.
>
>
> > There are several minor points related to the presentation that need to clarified / improved. Please see "Requested changes"
>
>
> ## Requested Changes
>
> ### Critical 1
>
> > Please include the results for the DHR-ARIMA baseline from Hewamalage et al., 2023 to the tables showcasing the forecasting performance (Section 4.2)
>
> We appreciate the reviewer's suggestion to include the DHR-ARIMA baseline from Hewamalage et al. (2023). Upon careful examination of their paper and corresponding scripts, we noticed that Hewamalage et al. did not explicitly state the input sequence length used for DHR-ARIMA. However, since they directly cite Informer results for comparison, we examined the Informer implementation and found that it uses an input sequence length of 336.
>
> In our experiments, the sequence length is consistently set to 96 across all benchmarks. Thus, for a fair and consistent comparison within our framework, we used a sequence length of 96 for evaluating DHR-ARIMA.
>
> We conducted experiments using both the R and Python implementations of DHR-ARIMA following (https://github.com/rakshitha123/TSForecasting). Our results indicate notably poor performance for DHR-ARIMA under our experimental conditions. For instance, on the ETTh1 dataset, our method achieves an MAE of 0.451, whereas DHR-ARIMA yields an MAE of 2.618.  This higher error aligns with the performance scale observed for methods such as Prophet, as reported in Table 2 of [1]. Similar patterns of poor performance were observed across other benchmark datasets we have tried.
>
>
>     [1] Cui Y, Xie J, Zheng K. Historical inertia: A neglected but powerful baseline for long sequence time-series forecasting[C]//Proceedings of the 30th ACM international conference on information & knowledge management. 2021: 2965-2969.

---

> > ### Author Response · Authors · 2025-03-07
> > **Rebuttal 2/2**
> >
> > ### Critical 2
> > > Why do the ablation experiments in Section 4.6 only consider 4 datasets (Weather, ETTh1, ETTm1, ILI) and omit the 4 datasets (ETTh2, ETTm2, ECL, Traffic) used in previous experiments?
> >
> > We thank the reviewer for this observation. In our ablation experiments, we initially focused on a subset of representative datasets (Weather, ETTh1, ETTm1, ILI) to clearly illustrate the effect of each design component. To address the concern, we have now conducted additional experiments on the remaining datasets (ETTh2, ETTm2, ECL, Traffic) and added these results to Table 5. We observe that the removal of any design component often leads to a decrease in performance.
> >
> >
> > ### Critical 3
> > > How is the ablation "w/o template" performed?
> >
> >
> > As discussed in Section 4.6 (Ablations), the "w/o template" ablation is performed by removing the designed template that formats the time series data, while still retaining the raw time series inputs to the LLM. This variant helps us isolate the effect of the template on mutual information extraction.
> >
> > ### Nice to have 4
> > > Section 3.1.1: How is the discriminator
> >  defined?
> >
> > As discussed in Section 3.1.1, the discriminator T
> > is implemented by feeding the positive and negative examples into a single-layer fully-connected network, after which we output the dot product of the resulting representations. To clarify further, we have now specified that the hidden dimension of this fully-connected network is set to 64.
> >
> > ### Nice to have 5
> > > Section 3.2: Please clarify the meaning of the prediction loss used in different tasks.
> >
> >
> > Thank you for requesting clarification on how we compute the prediction losses. Below is a concise summary, and we have updated Section 3.2 accordingly:
> >
> > - **Forecasting:**
> >   We take `[N, input_len, num_features]` as input and predict `[N, output_len, num_features]`. We compute the MSE/MAE for each sample (summing over the `output_len × num_features` dimension).
> >
> > - **Classification:**
> >   Each sample is associated with a class label. We compute the cross-entropy loss per sample.
> >
> > - **Imputation:**
> >   We randomly mask certain points in each sample, compute the MSE for those masked points for each sample.
> >
> > - **Anomaly Detection:**
> >   We adopt a reconstruction-based approach, computing the MSE for each sample between the original input and its reconstructed output.
> >
> >
> > ###  Nice to have 6
> > > Why are none of the related models described in Section 5.2 (e.g., Time-LLM, LLMTIME) are not considered in the evaluation? Just a description in the text would be sufficient.
> >
> >
> > To clarify:
> >
> > - **Time-LLM:** Its performance is compared in our work. As detailed in Section 4.2.2, the results are presented in Table 1 and Table 2.
> >
> > - **LLMTIME:** As stated in Section 5.1, *“Given that it is not a state-of-the-art method (worse than Fedformer) and primarily targets zero-shot forecasting, it has not been incorporated into our experimental framework.”* This rationale guided our decision to focus on models that demonstrate competitive performance in the forecasting task.
> >
> >
> >
> > ### Nice to have 7
> > > Given that the template used in this work only contains simple summary statistics of the time series and the original TS values, it would be interesting to investigate whether the LLM component is necessary at all. As an ablation, one could replace the LLM with an MLP that takes an input the appropriately normalized concatenated statistics rrom the template and outputs the embedding.
> >
> > Given that our template comprises only simple summary statistics of the time series and the original values, we investigated whether the LLM component is essential. For this ablation, we replaced the LLM with an MLP that accepts the normalized concatenated statistics (mean, max, median, lags) and outputs an embedding of identical dimension to the LLM.
> >
> > We conducted experiments on ETTh1 and ETTm1, with the following results:
> > - **ETTm1:** MSE increased from 0.401 to 0.420 (MAE from 0.409 to 0.425).
> > - **ETTh1:** MSE increased from 0.454 to 0.478 (MAE from 0.451 to 0.463).
> >
> > These results demonstrate that the LLM component is crucial in effectively capturing the mutual information from the template, thereby justifying its use in our approach. We have added the results in Appendix A.12.

---

> > > ### Author Response · Authors · 2025-03-13
> > > **Looking forward to your feedback**
> > >
> > > Thank you very much for your detailed review and constructive feedback. We have addressed each of your listed concerns as follows:
> > >
> > > 1. Revised the manuscript to highlight our commitment to exploring more diverse and robust datasets in future work.
> > > 2. Confirmed and clarified our baseline implementation, explaining the performance discrepancies observed in Figures 7 and 8.
> > > 3. Conducted additional experiments related to DHR-ARIMA.
> > > 4. Expanded the ablation studies in Section 4.6 to further validate the effectiveness of our LLM-TS method.
> > > 5. Provided clarification regarding how the ablation study without the template ("w/o template") was performed.
> > > 6. Clarified the discriminator used in our method.
> > > 7. Clarified the specific meanings of prediction loss across different tasks.
> > > 8. Clarified our comparisons with Time-LLM and LLMTIME.
> > > 9. Replaced the LLM with an MLP to empirically demonstrate the effectiveness of using LLM.
> > >
> > > If there is anything that remains unclear or requires further elaboration, please let us know. We look forward to your feedback and further suggestions.

---

> > > > ### Comment · Reviewer_zRNN · 2025-04-04
> > > > **Response from the reviewer**
> > > >
> > > > Thank you for the detailed response, it has addressed most of my concerns. I want to highlight that the provided setup with the artificially restricted context length for the ARIMA model is not a fair comparison, given the differences in the meaning of the "context length" for statistical and DL models. Limiting the `context_length` for the ARIMA will also prevent it from training on any data other than the last 96 observations, in contrast to DL models that use all the available data for training. Regardless of this point, since the response has addressed all of my other questions and concerns, I recommend acceptance for this work.

---

> > > > > ### Author Response · Authors · 2025-04-04
> > > > >
> > > > > We sincerely appreciate your thoughtful feedback and your support for acceptance. Thank you for highlighting the point about context length, and we will clarify this distinction in future revisions.

---

### Review · Reviewer_XMZc · 2025-02-24

**Summary Of Contributions:**

It introduces the LLM-TS Integrator framework, which fuses traditional time series (TS) models with the pattern recognition strengths of large language models (LLMs) by maximizing the mutual information between TS representations and their textual descriptions.

• It proposes a new sample reweighting module. This module assigns dual weights—one for the traditional prediction loss and another for the mutual information loss—optimizing sample importance via bi-level optimization, thereby enhancing overall information utilization.

• It validates the framework through extensive experiments across five key TS tasks (short-term and long-term forecasting, imputation, classification, and anomaly detection). The results show that LLM-TS Integrator consistently achieves state-of-the-art or comparable performance compared to both traditional TS methods and recent LLM-based approaches.

• It achieves these improvements while keeping the LLM component frozen (i.e., without fine-tuning), which minimizes additional computational costs and facilitates easy integration with existing TS models.

**Audience:**

Yes

**Claims And Evidence:**

Yes

**Requested Changes:**

The authors are required to include detailed time complexity analysis both theoretically and empirically.

**Strengths And Weaknesses:**

Pros:

1.The framework creatively fuses traditional time series models with LLM-derived insights. By maximizing the mutual information between numerical TS representations and their textual descriptions, it leverages the rigorous mathematical modeling of traditional approaches while tapping into the powerful pattern recognition capabilities of LLMs.


2.The dual-module approach—combining a mutual information module with a novel sample reweighting mechanism—allows the model to adaptively balance learning from prediction errors and informational alignment. This leads to improved performance across diverse TS tasks such as forecasting, imputation, classification, and anomaly detection.

3.By keeping the LLM component frozen, the framework avoids the computational overhead associated with fine-tuning large language models. This design choice makes the method more accessible and easier to integrate with existing TS models while still harnessing LLM insights.

4.The introduction of a sample reweighting module—where weights for prediction loss and mutual information loss are learned through bi-level optimization—ensures that each sample contributes effectively to the learning process. This adaptive mechanism helps the model focus on samples that need more learning, thereby improving overall model efficiency.

Cons:
1. The combined optimization of mutual information and prediction loss, especially with the bi-level optimization for sample reweighting, adds layers of complexity to training. Tuning the additional hyperparameters and ensuring stable convergence can be challenging. The authors have not provide detailed time complexity analysis both theoretically and empirically.

2.Accurately estimating mutual information between TS and textual representations is nontrivial. While the Jensen-Shannon estimator (and alternatives like MINE) is used, such estimators can be sensitive to the choice of architecture, learning rates, and data distribution. Misestimation might negatively affect the model’s performance.

3.The framework generates textual descriptions for time series data using a pre-defined template. The quality and informativeness of these generated texts are crucial. If the template does not capture the full complexity of the TS data, the resulting LLM-derived features might not be as beneficial.

---

> ### Author Response · Authors · 2025-03-07
> **Rebuttal**
>
> ## General reply
>
> Thank you for your thoughtful feedback, which has significantly enhanced our paper. In response, we have updated the manuscript with all changes clearly marked in red for your review.
>
> ## Cons:
>
> > 1. The combined optimization of mutual information and prediction loss, especially with the bi-level optimization for sample reweighting, adds layers of complexity to training. Tuning the additional hyperparameters and ensuring stable convergence can be challenging. The authors have not provide detailed time complexity analysis both theoretically and empirically.
>
> See Requested Changes
>
> > 2. Accurately estimating mutual information between TS and textual representations is nontrivial. While the Jensen-Shannon estimator (and alternatives like MINE) is used, such estimators can be sensitive to the choice of architecture, learning rates, and data distribution. Misestimation might negatively affect the model’s performance.
>
> Based on the ablation study results shown in Table 5, removing the mutual information component degrades the performance. For example, on the ETTh1 dataset, the MSE loss increases by almost 3\% when mutual information is excluded. These empirical results confirm that our approach is effective.
>
>
> > 3. The framework generates textual descriptions for time series data using a pre-defined template. The quality and informativeness of these generated texts are crucial. If the template does not capture the full complexity of the TS data, the resulting LLM-derived features might not be as beneficial.
>
> In Appendix A.5 "Template Variation", we present extensive experiments showing that performance remains quite consistent across various template modifications. These results confirm the robustness of our approach.
>
>
> ## Requested Changes:
> > The authors are required to include detailed time complexity analysis both theoretically and empirically.
>
> From a theoretical standpoint, let \(d_{\alpha}\) denote the dimension of the weighting network parameters and \(d_{\theta}\) denote the dimension of the prediction network parameters. Denote the computation cost of Eq. (6) as \(c(d_{\alpha}, d_{\theta})\). According to Section 4 of [1], the time complexity of the bi-level optimization is \(O(c(d_{\alpha}, d_{\theta}))\), with the primary computational cost arising from the Jacobian-vector product in Eq. (7). We have revised our paper to incorporate this theoretical discussion in Appendix A.6.
>
> Empirically, we have already detailed the time cost of each component for the ETTh1 and ETTm1 tasks using a batch size of 32 on a 32G V100 GPU in Table 10 of our original manuscript. The results are summarized in the table below:
>
> | Methods | Overall | TimesNet | Mutual Information | Sample Reweighting |
> |---------|---------|----------|--------------------|--------------------|
> | ETTh1   | 3.177   | 0.126    | 0.577              | 2.474              |
> | Weather | 5.563   | 0.436    | 1.094              | 4.033              |
>
> The training cost of our method is reasonable given that it achieves the best performance across most tasks. Furthermore, after training, the inference cost of our method is equivalent to that of TimesNet.
>
> [1] Franceschi L, Donini M, Frasconi P, et al. *Forward and reverse gradient-based hyperparameter optimization*. ICML 2017.

---

> > ### Author Response · Authors · 2025-03-13
> > **Looking forward to your feedback**
> >
> > Dear Reviewer XMZc,
> >
> > Thank you for your detailed review and constructive feedback. We have addressed all of your listed concerns, including:
> >
> > 1. Providing a detailed theoretical and empirical analysis of the time complexity for our LLM-TS method.
> > 2. Clarifying the effectiveness of the mutual information maximization module through empirical performance.
> > 3. Demonstrating the robustness of our method against template variation.
> >
> > Please let us know if any aspect remains unclear or requires further elaboration. We appreciate your valuable feedback and look forward to your response.

---

### Review · Reviewer_QjYM · 2025-02-27

**Summary Of Contributions:**

This paper presents a hybrid approach to time series analysis that couples the power of LLMs and that of tailored time series forecasting approaches based on mathematical modeling. In particular, the proposed approach suggests feeding a text prompt with time series data, its description, and some summary statistics to extract a hidden representation of this prompt from LLM. In parallel, a powerful TimesNet model is also fed with the same time series to produce its latent embedding. Both embeddings are then aligned using mutual information-based criterion with the strength of each embedding being determined with a weighting 1-layer fully connected network. The final representation is then used for such downstream tasks as forecasting, classification, anomaly detection, and imputation achieving strong results compared to existing baselines.

**Audience:**

Yes

**Claims And Evidence:**

Yes

**Requested Changes:**

1. I would greatly appreciate it if the authors addressed the concerns raised above (they are given in the order of priority). I would particularly like to understand how the proposed method (which I found novel and interesting, given that it doesn’t require the LLM during inference) compares to the field's current state, especially compared to foundation models (at least 1).
2. I think it will be quite useful to modify Figure 1 to show which parts are fine-tuned/frozen during training. Similarly, it will be great to illustrate here that the LLM is not used during inference but during training (or just before) only. Readers may find it appealing for practical applications.

**Strengths And Weaknesses:**

**Strengths**
1. The idea of using mutual information criterion to align the two embedding spaces is novel
2. The empirical results are good across 5 different tasks
3. Ablation studies provide interesting insights into the methods

**Weaknesses**
1. A more thorough comparison with baselines and discussion about the differences with TEST would be appreciated
It would be great to see how the embeddings aligned within the TEST model (based on contrastive learning) compared to those obtained by the authors. The two methods seem similar and understanding what brings a substantial advantage to the proposed approach would be an interesting contribution.
Similarly, the paper doesn’t use recent (universal) foundation models (Units, MOMENT, etc) that can all be applied to the considered tasks. It would be nice to have them or explain why the authors excluded them.
2. In A.6 the authors provide the efficiency analysis for inference speed but not for memory footprint.
I understand that for training, the embeddings from LLM can be obtained beforehand and the training can be done without loading LLM to the memory. However, extracting the embeddings from a large enough dataset can be very time-consuming and may incur a significant memory footprint. This gets even worse when dealing with multivariate datasets. I think a more extended discussion is necessary here.
3. A recent work [1] showed that LLM-based methods are not very useful in time series analysis: they can often be replaced with standalone models with better or similar accuracy. It will be great to discuss why in this particular case, LLMs seem to bring the advantage.

[1] Tan et al. Are Language Models Actually Useful for Time Series Forecasting? NeurIPS’24.

---

> ### Author Response · Authors · 2025-03-07
> **Rebuttal 1/2**
>
> ## General reply
>
> We greatly value your insightful feedback, which has significantly enhanced our paper. In light of your suggestions, we have updated the manuscript accordingly, with all modifications highlighted in red for your convenience.
>
> ## Weakness
>
> > 1. A more thorough comparison with baselines and discussion about the differences with TEST would be appreciated. It would be great to see how the embeddings aligned within the TEST model (based on contrastive learning) compared to those obtained by the authors. The two methods seem similar and understanding what brings a substantial advantage to the proposed approach would be an interesting contribution.
>
> In our original manuscript, we have already incorporated a comparison with TEST in the short-term forecasting task (Table 1), the long-term forecasting task (Table 2), and the classification task (Figure 3). Across these tasks, our method consistently outperforms TEST, which we attribute to our use of the TS model as the backbone that better captures underlying mathematical insights.
>
> Although the contrastive learning approach in TEST aligns representative embeddings with their corresponding TS embeddings—akin to our use of mutual information—the objectives of the two methods differ. TEST uses the aligned embeddings in an LLM backbone, a process we find to be less efficient. In contrast, our approach integrates the aligned embeddings directly within the TS backbone, leading to more effective performance.
>
> A direct embedding comparison between TEST and our model is challenging, given that the embeddings are optimized for distinct objectives and vary in dimensionality. Thus, we believe our current evaluations sufficiently demonstrate the advantages and distinctiveness of our approach.
>
>
>
> >  Similarly, the paper doesn’t use recent (universal) foundation models (Units, MOMENT, etc) that can all be applied to the considered tasks. It would be nice to have them or explain why the authors excluded them.
>
> We thank the reviewer for highlighting additional baseline methods such as MOMENT and UniTS. We have carefully considered these universal foundation models for our evaluations:
>
> - MOMENT is pre-trained specifically for sequences of fixed length (T=512), which significantly exceeds the sequence length used in our setting (T=96). Therefore, a direct and fair comparison with MOMENT under our experimental conditions is not feasible.
>
> - Regarding UniTS, although its reported performance appears strong in certain tasks, our experimentation using their publicly available code resulted in notably lower performance than our method, aligning with the observations from other users documented at https://github.com/mims-harvard/UniTS/issues/27 (e.g., anomaly detection performance on the SMD dataset).
>
> Furthermore, we emphasize that our objective is not to achieve state-of-the-art performance across benchmarks but rather to propose and validate the effectiveness of our LLM-TS integrator. We have cited both UniTS and MOMENT in our manuscript and included relevant discussions in Section 5.1: "\cite{gao2024units, goswami2024moment} also explore time series foundation models by pre-training large models on extensive time series datasets."
>
>
> > 2. In A.6 the authors provide the efficiency analysis for inference speed but not for memory footprint. I understand that for training, the embeddings from LLM can be obtained beforehand and the training can be done without loading LLM to the memory. However, extracting the embeddings from a large enough dataset can be very time-consuming and may incur a significant memory footprint. This gets even worse when dealing with multivariate datasets. I think a more extended discussion is necessary here.
>
> We thank the reviewer for highlighting the need for an extended discussion regarding the memory footprint associated with obtaining embeddings from LLMs. In the original manuscript (Appendix A.6), we already mentioned the efficiency of our embedding extraction process: "obtaining the embeddings for the ETTh1 dataset using the llama-3b model on an A100 GPU takes approximately 1 hour." We consider this to be quite efficient given the scale of the model and dataset.
>
> However, we acknowledge that embedding extraction may become more time-consuming and memory-intensive with significantly larger or multivariate datasets. In such scenarios, strategies such as prioritizing difficult or representative data points could be employed to manage computational costs effectively. Additionally, it is important to note that inference efficiency, a primary practical concern, is the same as the efficient backbone models like TimesNet. We have expanded our discussion accordingly in Appendix A.6 to clarify these points.

---

> > ### Author Response · Authors · 2025-03-07
> > **Rebuttal 2/2**
> >
> > > 3. A recent work [1] showed that LLM-based methods are not very useful in time series analysis: they can often be replaced with standalone models with better or similar accuracy. It will be great to discuss why in this particular case, LLMs seem to bring the advantage.
> >
> > We appreciate the reviewer pointing out the recent work by Tan et al. [1], which questions the utility of LLM-based methods in time series analysis, primarily focusing on cases where LLMs are employed directly as prediction backbones. In contrast, our approach utilizes LLMs in an auxiliary capacity, leveraging their rich representational power to enhance a traditional time-series backbone rather than relying on them directly for predictions.
> >
> > To specifically address this point, we have conducted an ablation study (detailed in Appendix A.12) where the LLM component was replaced with a standard MLP. The results clearly indicate degraded performance when the LLM embeddings were removed, confirming the beneficial contribution of the LLM component.
> >
> > We have cited Tan et al. [1] explicitly in our revised manuscript within Appendix A.12, clarifying our position and stating that our method differs significantly in how it utilizes LLM embeddings: "Tan et al. [1] demonstrate that using LLMs as prediction backbones may not yield substantial improvements; however, our results highlight the effectiveness of LLM embeddings when integrated as auxiliary information within traditional time-series models."
> >
> > [1] Tan et al. Are Language Models Actually Useful for Time Series Forecasting? NeurIPS’24.
> >
> >
> > ## Requested Changes:
> > > 1. I would greatly appreciate it if the authors addressed the concerns raised above (they are given in the order of priority). I would particularly like to understand how the proposed method (which I found novel and interesting, given that it doesn’t require the LLM during inference) compares to the field's current state, especially compared to foundation models (at least 1).
> >
> > Thank you for your valuable suggestions. We have addressed each point in detail.
> >
> >
> > > 2. I think it will be quite useful to modify Figure 1 to show which parts are fine-tuned/frozen during training. Similarly, it will be great to illustrate here that the LLM is not used during inference but during training (or just before) only. Readers may find it appealing for practical applications.
> >
> > In our original Figure 1, we used colors to distinguish the trainable components (yellow) from the frozen component (blue): specifically, the Backbone (TimesNet), Mutual Information Maximization, and Weighting Net are trainable, while the LLM is frozen.
> >
> > We are currently improving the figure to make these distinctions clearer and to explicitly illustrate that the LLM is not used during inference.  Additionally, we have updated the figure caption in the revised manuscript to clarify: "The LLM is utilized solely during the training phase and is not required during inference."

---

> > > ### Author Response · Authors · 2025-03-11
> > > **Figure Updated**
> > >
> > > Dear Reviewer QjYM,
> > >
> > > We have revised the figure to more clearly delineate the training and inference phases. In the updated version, it is explicitly shown that the LLM is employed solely during training and omitted from the inference process.

---

> > > > ### Author Response · Authors · 2025-03-13
> > > > **Looking forward to your feedback**
> > > >
> > > > Dear Reviewer QjYM,
> > > >
> > > > Thank you for your detailed review and constructive feedback. We have carefully addressed each of your main concerns, including:
> > > >
> > > > 1. Clarifying our comparisons with other methods, such as TEST.
> > > > 2. Providing an efficiency analysis specifically addressing memory footprint.
> > > > 3. Conducting an ablation study to demonstrate the effectiveness of LLM in its auxiliary role within our framework.
> > > > 4. Modifying Figure 1 to clearly indicate that the LLM is not utilized during inference.
> > > >
> > > > Please let us know if any aspect remains unclear or requires further elaboration. We appreciate your guidance and look forward to your feedback.

---

> > > > > ### Comment · Reviewer_QjYM · 2025-03-13
> > > > > **thank you**
> > > > >
> > > > > I would like to thank the authors for their reply, which cleared up most of my concerns. Also, I think the authors misunderstood to some extent [1] as it shows that when LLM's frozen backbone is used as part of the model that fine-tunes or trains some additional layers, then LLM's backbone becomes not very useful. Similarly, it is not very meaningful to replace LLM with an MLP in Appendix A.12 as LLM is a transformer-based model and [1] shows that it should be replaced either with an attention module or a single transformer block, not with an MLP. I believe that this part is still a bit unclear but otherwise it is fine

---

> > > > > > ### Author Response · Authors · 2025-03-13
> > > > > > **Thank you**
> > > > > >
> > > > > > We thank the reviewer for their comments, which helped clarify important aspects of our work. We acknowledge that the reviewer highlights a possible misunderstanding regarding reference [1]. Specifically, [1] demonstrates that a frozen LLM backbone has limited utility when directly integrated into models trained with additional layers for downstream prediction. However, our approach (LLM-TS) differs significantly: we leverage LLM features as a supplementary enhancement to the traditional time series (TS) model representation, rather than using them directly for prediction at inference. This critical distinction indicates that, while directly utilizing LLM features in predictive tasks may have limitations, these features can still benefit performance when integrated as an augmentation to traditional TS modeling.
> > > > > >
> > > > > > Additionally, following the reviewer's suggestion, we experimented with replacing the MLP with a single-layer self-attention module. Still, this adjustment led to suboptimal results compared with LLM. Specifically, on the ETTm1 dataset, the MSE increased from $0.401$ to $0.423$, and the MAE increased from $0.409$ to $0.419$. Similarly, on the ETTh1 dataset, the MSE increased from $0.454$ to $0.475$, and the MAE from $0.451$ to $0.461$. We have included these details in Appendix A.12 of the revised manuscript to provide further clarity.

---

### Decision · Action_Editor_wsA4 · 2025-04-13

**Recommendation:** Accept as is

**Comment:**

The paper tackles a very timely problem which consists of leveraging the expressiveness of time series models with the power of large language models. The major contribution of the paper can be summarized as follows: the same time series is fed to both the time series model and the LLM to generate emdedding based representation which can then be used for multiple downstream tasks. The idea is simple yet effective and the strong experimental results across 5 different time series tasks show that the proposed framework can capture the embedded knowledge within the time series.

The provided rebuttal was appreciated well by the reviewers and there was a clear consensus among the reviewers for an accept. I would like to ask the authors to consider the reviews and their provided rebuttal into account when preparing the camera ready version of the manuscript.

I would like to congratulate the authors on the nice work and the acceptance.

**Audience:**

Yes. A broad set of audience will be interested in this work.

**Claims And Evidence:**

Yes, the claims in the submission seem accurate and are supoported by clear empirical evidence.